# Shared Stochastic Gaussian Process Latent Variable Models: A Multi-modal Generative Model for Quasar Spectra

**Vidhi Lalchand**                                                                *vidrl@mit.edu*
*Eric and Wendy Schmidt Center*
*Broad Institute of MIT and Harvard; MIT*

**Anna-Christina Eilers**                                                          *eilers@mit.edu*
*MIT Kavli Institute for Astrophysics and Space Research*

**Reviewed on OpenReview:** *https://shorturl.at/SUjLP*

## Abstract

This work proposes a scalable probabilistic latent variable model based on Gaussian processes (Lawrence, 2004) in the context of multiple observation spaces. We focus on an application in astrophysics where it is typical for data sets to contain both observed spectral features as well as scientific properties of astrophysical objects such as galaxies or exoplanets. In our application, we study the spectra of very luminous galaxies known as quasars, and their properties, such as the mass of their central supermassive black hole, their accretion rate and their luminosity, and hence, there can be multiple observation spaces. A single data point is then characterised by different classes of observations, which may have different likelihoods. Our proposed model extends the baseline stochastic variational Gaussian process latent variable model (GPLVM) (Lalchand et al., 2022) to this setting, proposing a seamless generative model where the quasar spectra and scientific labels can be generated *simultaneously* when modelled with a shared latent space acting as input to different sets of Gaussian process decoders, one for each observation space. In addition, this framework allows training in the missing data setting where a large number of dimensions per data point may be unknown or unobserved. We demonstrate high-fidelity reconstructions of the spectra and the scientific labels during test-time inference and briefly discuss the scientific interpretations of the results along with the significance of such a generative model. The code for this work is available at `https://github.com/vr308/Quasar-GPLVM`.

## 1 Introduction

Many challenges in the contemporary physical sciences arise from the analysis of large-scale, noisy, heteroscedastic, and high-dimensional datasets (Clarke et al., 2016). Hence, there is increasing consensus that addressing the challenges posed by large-scale data in the experimental pipelines for discovery, forecasting, and prediction warrant scalable machine learning. Modern experiments in physics, chemistry, and astronomy are capable of producing extremely complex high-dimensional data, but it is not just the sheer volume of the data, but also the *velocity*, referring to the rate of production (Carleo et al., 2019) of data, which poses an additional challenge. Further, an additional axis is the variety or heterogeneity inherent in scientific datasets in the form of multiple outputs or observation spaces. For instance, images (pixels) can be accompanied with attributes like illumination, pose, and resolution. In addition, some of these attributes or observed features are often unknown or unobserved, resulting in incomplete data sets with missing components. The multiple output setting is inadequately addressed in the probabilistic machine learning literature; in this work we tackle precisely this setting proposing a multi-output scalable probabilistic model for a scientific application in astronomy.

A preponderance of literature in unsupervised learning focuses on the setting of a single large-scale homogeneous dataset, for instance, images, text, speech, or continuous numerical values. Further, most parametric models

also assume a complete dataset where each dimension of every data point is observed. Real-world data are often only partially observed and in some cases yield very sparse data matrices with majority of the dimensions missing (Corduneanu & Jaakkola, 2012). Robust unsupervised learning in these settings is a challenge and one needs to account for sensible epistemic uncertainties. In the running application we consider in this work, handling missing data during training is crucial, as spectral pixels in real observations can be missing due to absorption features in the Earth's atmosphere or different wavelength coverage of the spectra when observed with different telescopes or spectrographs.

Dimensionality reduction is a standard precursor to further analysis in scientific datasets, as the intrinsic dimensionality of the data might actually be quite low. It is usually possible to summarise the key axis of variation in the data with very few dimensions (Facco et al., 2017) side-stepping the curse of dimensionality and facilitating further downstream analysis. Projection techniques like PCA, multidimensional scaling (MDS) and independent component analysis (ICA) utilise eigenvalue decomposition (Murphy, 2012) while other non-linear techniques like t-SNE (van der Maaten & Hinton, 2008), SNE (van der Maaten, 2009) and UMAP (McInnes et al., 2020) construct a probability distribution on high dimensional points and replicate a similar distribution on low-dimensional points iteratively using the KL divergence. These methods, however, are not generative in their traditional incarnation. They cannot be used to generate instances of the high-dimensional data points; hence, they are not very useful in many astrophysical settings where the ability to generate points in high-dimensional data space is crucial.

Generative latent variable models supplement traditional dimensionality reduction techniques as they offer the simultaneous benefits of a probabilistic interpretation and data generation while learning a faithful embedding of the high-dimensional training data in low-dimensional latent space. A generative probabilistic framework like the GPLVM (Lawrence, 2004) works by optimising the parameters of a Gaussian process *decoder* from a low dimensional latent space ($Z \in \mathbb{R}^{N \times Q}$) to high-dimensional data space ($X \in \mathbb{R}^{N \times D}$) such that $Q \ll D$ and points close in latent space are nearby in data space. Since the decoder is a non-parametric Gaussian process, the kernel function controls the inductive biases of the function mapping like smoothness and periodicity. There typically is no encoder mapping hence, these models are also called Gaussian process decoders. It is possible to additionally incorporate a back-constraint or an encoder which maps from the data to latent space putting GPLVMs on the same footing as variational auto-encoders (VAEs; Lawrence & Quiñonero Candela, 2006; Bui & Turner, 2015). This amortises the cost of variational inference in very large-scale datasets but we don't employ this setting as it is not straightforwardly applicable to missing data contexts.

This work proposes a novel formulation of the GPLVM based on the idea of a shared latent space. The earlier work by Ek (2009) was the first to propose the idea of a shared data generation process but precluded truly scalable inference due to the standard $\mathcal{O}(N^3)$ scaling. We extend this framework in two important ways. First, we show that the shared GPLVM is compatible with stochastic variational inference (SVI) (Hoffman et al., 2013) where we derive a joint evidence lower bound which factorises across multiple observation spaces due to conditional independence but share predictive strength though inducing locations and latent variables. Secondly, we train the entire model in presence of missing dimensions in one or both of the observation spaces.

We demonstrate this scalable model in an astrophysical application using data of quasars. Quasars are the most luminous galaxies in the universe, powered by accretion onto a central supermassive black hole (SMBH) with millions to billions of solar masses in size. Understanding the formation, growth, and evolution across cosmic time of quasars and their SMBHs is one of the major goals of observational cosmology today. To this end, precise measurements of the physical properties of quasars are crucial, but they typically demand very expensive and time-intensive observations, as multiple epochs are needed to accurately determine, for instance, the quasar's SMBH mass. The high-dimensional data used in this work contains $\sim 22,000$ quasars from the Sloan Digital Sky Survey (SDSS; Lyke et al., 2020; Wu & Shen, 2022) with high-quality spectral information (binned to 590 spectral dimensions/pixels) along with four scientific labels per quasar, i.e. their black hole mass, luminosity, redshift and so-called Eddington ratio – a measure of the quasar's accretion rate. By modelling the spectra and scientific labels through a generative model acting on a shared latent space we aim to reason about the physical properties of the quasars just through its "single-epoch" spectral information, thus circumventing the time-intensive multi-epoch observations. Earlier work on applying probabilistic generative modelling using a GPLVM to high-dimensional quasar spectra (Eilers et al., 2022)

have been constrained on scalability and examine less than 50 astronomical objects. We demonstrate our framework on datasets over 400× larger. We summarise our key contributions below:

**Contributions** We propose a probabilistic generative framework called the *Shared stochastic GPLVM* which is designed for use cases with multiple outputs/observation spaces. We seamlessly account for missing dimensions both at training and test time due to the probabilistic nature of the model. We demonstrate through astrophysical experiments that we can reconstruct previously unseen/test spectral pixels to a high degree of fidelity, interpolate missing or unobserved spectral regions and predict scientific labels. Crucially, we demonstrate that it is possible to share predictive strength by learning a common latent variable space $Z$ across multiple-outputs $(X, Y)$ where $Y \in \mathbb{R}^{N \times L}$ is an additional view of the data, we refer to this as an additional observation space with $L$ dimensions. In this way we indirectly model the relationships and correlation structure between the different observation spaces. We demonstrate this concretely with an experiment where we generate/predict all the scientific attributes at test-time by using latent variables ($Z$) only informed by the quasar spectra ($X$), we can denote this cross-modal prediction as $X \rightarrow Z \rightarrow Y$. To the best of our knowledge, predicting multiple outputs with stochastic variational GPs for scalability is methodologically novel. Further, this is the first demonstration of a scalable probabilistic latent variable model in astrophysical settings. In Section 10 we discuss how the development of these unsupervised learning frameworks can instigate novel insights into some of the major open questions in astronomy today.

## 2 Stochastic Variational GPLVM with a Shared latent space

In this section, we first describe background on the stochastic variational GPLVM (Lalchand et al., 2022). We then develop the idea of a shared latent space and inducing points within the stochastic variational GPLVM framework. The fundamental contribution of this work is to develop an inference scheme to show that a shared latent space does not preclude scalable inference through SVI.

### 2.1 SV-GPLVM: Stochastic Variational GPLVM

In the traditional formulation underlying GPLVMs we have a training set comprising of $N$ $D$-dimensional real-valued observations $X \equiv \{\boldsymbol{x}_n\}_{n=1}^N \in \mathbb{R}^{N \times D}$. These data are associated with $N$ $Q$-dimensional latent variables, $Z \equiv \{\boldsymbol{z}_n\}_{n=1}^N \in \mathbb{R}^{N \times Q}$ where $Q \ll D$ provides dimensionality reduction (Lawrence, 2004). The forward mapping ($Z \longrightarrow X$) is governed by GPs independently defined across dimensions $D$. The sparse GP formulation describing the data is as follows:

$$
\begin{aligned}
p(Z) &= \prod_{n=1}^N \mathcal{N}(\boldsymbol{z}_n; \boldsymbol{0}, \mathbb{I}_Q), \\
p(F|U, Z, \theta) &= \prod_{d=1}^D \mathcal{N}(\boldsymbol{f}_d; K_{nm} K_{mm}^{-1} \boldsymbol{u}_d, Q_{nn}), \\
p(X|F, Z) &= \prod_{n=1}^N \prod_{d=1}^D \mathcal{N}(x_{n,d}; f_d(\boldsymbol{z}_n), \sigma_x^2),
\end{aligned}
\tag{1}
$$

where $Q_{nn} = K_{nn} - K_{nm} K_{mm}^{-1} K_{mn}$ is $N \times N$, $F \equiv \{\boldsymbol{f}_d\}_{d=1}^D$ where $\boldsymbol{f}_d \in \mathbb{R}^N$, $U \equiv \{\boldsymbol{u}_d\}_{d=1}^D$ where $\boldsymbol{u}_d \in \mathbb{R}^M$ and $\boldsymbol{x}_d$ is the $d^{th}$ column of $X$. $K_{nn}$ is the $N \times N$ covariance matrix corresponding to a user-chosen positive-definite kernel function $k_\theta(\boldsymbol{z}, \boldsymbol{z}')$ evaluated at latent points $\{\boldsymbol{x}_n\}_{n=1}^N$ and parametrised by shared hyperparameters $\theta$. Inducing variables per dimension $\{\boldsymbol{u}_d\}_{d=1}^D$ are distributed with a GP prior $\boldsymbol{u}_d|\tilde{Z} \sim \mathcal{N}(\boldsymbol{u}_d; \boldsymbol{0}, K_{mm})$ (where $K_{mm}$ is $M \times M$) computed on inducing input locations $[\tilde{\boldsymbol{z}}_1, \ldots \tilde{\boldsymbol{z}}_M]^T \equiv \tilde{Z} \in \mathbb{R}^{M \times Q}$ which live in latent space with $Z$ and have dimensionality $Q$ (matching $\boldsymbol{z}_n$). In addition, $K_{nm}$ is the $N \times M$ cross-covariance computed on the latents $\boldsymbol{z}_n$ and the inducing locations $\tilde{\boldsymbol{z}}_m$.

The core idea of Stochastic Variational Inference (SVI) applied to sparse variational Gaussian processes (GPs), as introduced by Hensman et al. (2013), is that the inducing variables $\boldsymbol{u}_d$ can be variationally marginalized. This is achieved by learning their variational distribution $q(\boldsymbol{u}_d) \sim \mathcal{N}(\boldsymbol{m}_d, S_d)$ using stochastic gradient

methods. Crucially, this approach retains an uncollapsed representation of $\boldsymbol{u}_d$ throughout the optimization process. While Hensman et al. (2013) proposed SVI for GP regression, Lalchand et al. (2022) extended this work to GPLVMs where the inputs $Z$ to the GPs are unobserved and each dimension of the high-dimensional output space $\boldsymbol{x}_d$ is modelled by an independent GP $f_d$ with shared kernel hyperparameters.

Succinctly, posterior inference entails minimising the KL-divergence between the posterior over unknowns $p(F, U, Z|X)$ and the variational approximation $q(F, U, Z)$. Following Lalchand et al. (2022) when the variational approximation admits the factorisation below as in Titsias (2009):

$$q(F, U, Z) \approx p(F|U, Z)q(U)q(Z) = \prod_{d=1}^{D} p(\boldsymbol{f}_d|\boldsymbol{u}_d, Z)q(\boldsymbol{u}_d) \prod_{n=1}^{N} q(\boldsymbol{z}_n) \tag{2}$$

the evidence lower bound (ELBO) can be derived as,

$$\log p(X) \geq \langle \log p(X|F, Z) \rangle_{q(.)} - \mathrm{KL}(q(Z)||p(Z)) - \mathrm{KL}(q(U)||p(U)), \tag{3}$$

where $q(.)$ denotes the variational approximation as in Eq. (2). For brevity, we suppress the conditioning over the inducing inputs $\tilde{Z}$ in the prior $p(U)$ and the kernel hyperparameters $\theta$ in $p(F|U, Z)$.

If we choose to optimize the latent variables $Z$ as point estimates rather than variationally integrate them out (basically we do not introduce $q(Z)$) we end up with the following simplification,

$$p(X) \geq \int p(F|U, Z)q(U) \log \frac{p(X|F, Z)p(U)p(Z)}{q(U)} dF dU = \mathcal{L}_x \tag{4}$$

Re-writing the lower bound as a sum of terms across data points $N$ and outputs/dimensions $D$ we get,

$$\mathcal{L}_x = \sum_{n,d} \langle \log p(x_{n,d}|\boldsymbol{f}_d, \boldsymbol{z}_n, \sigma_x^2) \rangle_{p(\boldsymbol{f}_d|\boldsymbol{u}_d, Z)q(\boldsymbol{u}_d)} - \sum_d \mathrm{KL}(q(\boldsymbol{u}_d)||p(\boldsymbol{u}_d|\tilde{Z})) + \sum_n \log p(\boldsymbol{z}_n) \tag{5}$$

Latent point estimates $\{\boldsymbol{z}_n\}_{n=1}^{N}$ can be learnt along with $\theta$ and variational parameters $(\tilde{Z}, \boldsymbol{m}_d, S_d)$ by taking gradients of the ELBO in Eq. (5). However, an important constraint is that this formulation assumes a single-kernel matrix (single set of kernel hyperparameters) underlying all the independent GPs $D$. In the next section, we introduce the concept of an additional observation space with dimensions $L$, which can be modelled using an independent stack of Gaussian processes (GPs), $f_l$. These GPs are equipped to learn their own set of hyperparameters for added flexibility while sharing the latent embedding $Z$ and inducing inputs $\tilde{Z}$ to capture correlations across the different output spaces.

## 2.2 Shared joint variational lower bound

In the astrophysical application we focus on in this work we have two observation spaces corresponding to $N$ quasars. We denote the quasar spectra (pixels) with the matrix $X \in \mathbb{R}^{N \times D}$ and the scientific labels corresponding to the $N$ objects with $Y \in \mathbb{R}^{N \times L}$. The GPLVM construction models each column (pixel dimension and label dimension) with an independent GP, with the GPs corresponding to the pixel dimensions $\{f_d\}_{d=1}^{D}$ and label dimensions $\{f_l\}_{l=D+1}^{D+L}$ modelled with their own independent kernels and kernel hyperparameters, $\theta_x$ and $\theta_y$. Within each observation space the kernel hyperparameters are shared, so we learn two sets of hyperparameters corresponding to two observation spaces.

$$f_d \sim \mathcal{GP}(0, k_{\theta_x}) \qquad f_l \sim \mathcal{GP}(0, k_{\theta_y}) \tag{6}$$

The priors over the function values are given by,

$$p(\boldsymbol{f}_d|\theta_x) = \mathcal{N}(\mathbf{0}, K_{nn}^{(d)}) \tag{7}$$

$$p(\boldsymbol{f}_l|\theta_y) = \mathcal{N}(\mathbf{0}, K_{nn}^{(l)}) \tag{8}$$

where $K_{nn}^{(d)}$ and $K_{nn}^{(l)}$ denote the $N \times N$ kernel matrices which rely on their own set of hyperparameters. The two observation spaces also yield two data likelihoods given by,

$$p(X|\boldsymbol{f}_{1:D}, Z) = \prod_{n=1}^{N} p(\boldsymbol{x}_n|\boldsymbol{f}_d, Z) = \prod_{n=1}^{N}\prod_{d=1}^{D} \mathcal{N}(x_{n,d}; f_d(\boldsymbol{z}_n), \sigma_x^2) \tag{9}$$

$$p(Y|\boldsymbol{f}_{D+1:D+L}, Z) = \prod_{n=1}^{N} p(\boldsymbol{y}_n|\boldsymbol{f}_l, Z) = \prod_{n=1}^{N}\prod_{l=D+1}^{D+L} \mathcal{N}(y_{n,l}; f_l(\boldsymbol{z}_n), \sigma_y^2) \tag{10}$$

In the absence of sparsity the log-marginal likelihood of the joint model compartmentalises nicely due to the assumed factorisation in the likelihoods. We marginalise out the latent function values $f_{1:D}$ and $f_{D+1:D+L}$ per dimension,

$$p(X, Y|\theta_x, \theta_y, Z) = \int_{1:D} \int_{D+1:D+L} p(X|\boldsymbol{f}_{1:D}, Z)p(Y|\boldsymbol{f}_{D+1:D+L}, Z)p(\boldsymbol{f}_d|\theta_x)p(\boldsymbol{f}_l|\theta_y)d\boldsymbol{f}_{1:D}d\boldsymbol{f}_{D+1:D+L} \tag{11}$$

$$= \int_{1:D} p(X|\boldsymbol{f}_{1:D}, Z)p(\boldsymbol{f}_d|\theta_x)d\boldsymbol{f}_{1:D} \int_{D+1:D+L} p(Y|\boldsymbol{f}_{D+1:D+L}, Z)p(\boldsymbol{f}_l|\theta_y)d\boldsymbol{f}_{D+1:D+L} \tag{12}$$

$$= \prod_{d=1}^{D} p(\boldsymbol{x}_d|\theta_x, Z) \prod_{l=D+1}^{D+L} p(\boldsymbol{y}_l|\theta_y, Z) = \prod_{d=1}^{D} \mathcal{N}(\boldsymbol{0}, K_{nn}^{(d)} + \sigma_x^2) \prod_{l=D+1}^{D+L} \mathcal{N}(\boldsymbol{0}, K_{nn}^{(l)} + \sigma_y^2) \tag{13}$$

where $\boldsymbol{x}_d$ and $\boldsymbol{y}_l$ denote a single column/dimension of the observation spaces $X$ and $Y$. The log marginal likelihood objective is then given by the following,

$$\log p(X, Y|\theta_x, \theta_y, Z) = \sum_{d=1}^{D} \log p(\boldsymbol{x}_d|\theta_x, Z) + \sum_{l=D+1}^{D+L} \log p(\boldsymbol{y}_l|\theta_y, Z) \tag{14}$$

In order to induce sparsity we introduce inducing variables $\boldsymbol{u}_d$ and $\boldsymbol{u}_l$ for each individual dimension in the observation spaces; however, they are underpinned by shared inducing inputs $\tilde{Z}$ which live in the shared latent space $Z$ and share the same dimensionality, $Q$. With sparse GPs each of the terms in the decomposition above can be bounded by $\mathcal{L}_x$, while the inducing points $\tilde{Z}$ can be shared between the terms yielding the joint evidence lower bound.

$$\log p(X, Y|\theta_x, \theta_y, Z) \geq \sum_{n,d} \langle \log p(x_{n,d}|\boldsymbol{f}_d, \boldsymbol{z}_n, \sigma_x^2) \rangle_{p(\boldsymbol{f}_d|\boldsymbol{u}_d, Z)q(\boldsymbol{u}_d)} - \sum_{d} \mathrm{KL}(q(\boldsymbol{u}_d)||p(\boldsymbol{u}_d)) \tag{15}$$

$$+ \sum_{n,l} \langle \log p(y_{n,l}|\boldsymbol{f}_l, \boldsymbol{z}_n, \sigma_y^2) \rangle_{p(\boldsymbol{f}_l|\boldsymbol{u}_l, Z)q(\boldsymbol{u}_l)} - \sum_{l} \mathrm{KL}(q(\boldsymbol{u}_l)||p(\boldsymbol{u}_l)) + \log p(Z)$$

Overall we optimise the shared variational lower bound w.r.t kernel hyperparameters for the two groups of GPs, $\theta_x$ and $\theta_y$, variational parameters $\{\boldsymbol{m}_d, S_d\}_{d=1}^{D}$ and $\{\boldsymbol{m}_l, S_l\}_{l=D+1}^{D+L}$, and a single set of shared latent point estimates $Z$ and inducing inputs $\tilde{Z}$. We include the full training algorithm in Algorithm 2.

## 3   Predictions & Reconstructions

High-dimensional points can arrive in different formats for the test data, either we observe both modalities $\{\boldsymbol{x}^*, \boldsymbol{y}^*\}$ where $\boldsymbol{x}^* = [x_1^*, \ldots, x_d^*]^T$ and similarly for $\boldsymbol{y}^*$ or only one of the modalities with the other missing, i.e. $\{\boldsymbol{x}^*\}$ or $\{\boldsymbol{y}^*\}$ only. The prediction exercise then involves inferring the latent $\boldsymbol{z}^*$ corresponding to the unseen test point.

Since the GPLVM is a decoder only model, we cannot obtain the latent embedding $\boldsymbol{z}^*$ deterministically; instead we re-optimize the ELBO with the additional test data point $(\boldsymbol{x}^*, \boldsymbol{y}^*)$ while keeping all the global and model hyperparameters frozen at their trained values. Note that since the ELBO factorises across data points, $\mathcal{L}(\{\boldsymbol{x}_n, \boldsymbol{y}_n\}_{n=1}^{N}, \boldsymbol{x}^*, \boldsymbol{y}^*) = \sum_{n=1}^{N+1}\sum_{h=1}^{D+L} \mathcal{L}_{n,h}$, the gradients to derive the new latent point $\boldsymbol{z}^*$ only depend

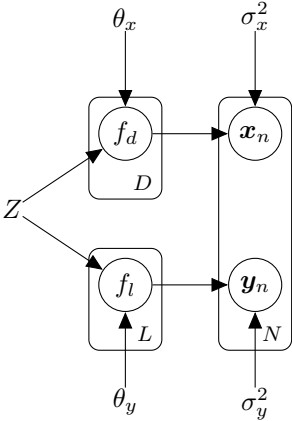

Figure 1: The graphical model of the shared GPLVM with two sets of independent GPs and their respective hyperparameter sets.

on the respective component terms connected to the data point. In the event of a missing dimension $d^*$ for a new data point $\boldsymbol{x}^*$, the augmented ELBO for the spectra dimensions can be written as follows:

$$\mathcal{L}_x^* \leftarrow \mathcal{L}_x + \log p(\boldsymbol{z}^*) + \sum_{d \neq d^*} \langle p(\boldsymbol{x}_d^*, | \boldsymbol{f}_d, \boldsymbol{z}^*, \sigma_x^2) \rangle_{q(.)} \tag{16}$$

Once we infer $\boldsymbol{z}^*$ we can compute the full reconstruction distributions (we may be interested in this if the test data point $\boldsymbol{y}^*$ had any missing dimensions, for example $\boldsymbol{y}^* = [y_1^*, y_2^*, ?, ?, y_5^*, \ldots, y_l^*]^T$) which are just the GP posterior predictive for each column or dimension, without loss of generality, for dimension $y_l^*$:

$$\begin{aligned} p(y_l^* | \boldsymbol{z}^*) &= \int p(y_l^* | \boldsymbol{f}_l, \boldsymbol{u}_l, \boldsymbol{z}^*) p(\boldsymbol{f}_l, \boldsymbol{u}_l | \boldsymbol{y}_l) d\boldsymbol{f}_l d\boldsymbol{u}_l \\ &= \int p(y_l^* | \boldsymbol{f}_l, \boldsymbol{u}_l, \boldsymbol{z}^*) p(\boldsymbol{f}_l | \boldsymbol{u}_l) q(\boldsymbol{u}_l) d\boldsymbol{f}_l d\boldsymbol{u}_l \\ &= \int p(y_l^* | \boldsymbol{u}_l, \boldsymbol{z}^*) q(\boldsymbol{u}_l) d\boldsymbol{u}_l \end{aligned} \tag{17}$$

where $p(y_l^* | \boldsymbol{u}_l, \boldsymbol{z}^*) = \mathcal{N}(K_{*m} K_{mm}^{-1} \boldsymbol{u}_l, K_{**} - K_{*m} K_{mm}^{-1} K_{m*} + \sigma_y^2)$ and $q(\boldsymbol{u}_l) = \mathcal{N}(\boldsymbol{m}_l^*, S_l^*)$ refers to the optimised variational distribution. The final integral is tractable and gives the following form:

$$p(y_l^* | \boldsymbol{z}^*) = \mathcal{N}(K_{*m} K_{mm}^{-1} \boldsymbol{m}_l^*, K_{*m} K_{mm}^{-1} (S_l^* - K_{mm}) K_{mm}^{-1} K_{m*} + \sigma_y^2) \tag{18}$$

and similarly for any $x_d^*$. For cross-modal reconstruction (where we only observe one modality of the test data) the latent $\boldsymbol{z}^*$ acts as the information bottleneck, hence, the same posterior predictive distributions can be derived, $\boldsymbol{x}^* \to \boldsymbol{z}^* \to p(y_l^* | \boldsymbol{z}^*) \quad \forall l = D + 1, ..., D + L$.

## 4 Schematic of the Model

In Fig. 2 we present a schematic of the model architecture with two observation spaces $(X, Y)$, the corresponding stacks of individual GPs $\{f_d\}$ and $\{f_l\}$ which model the individual columns of the spectra $X$ and scientific attributes $Y$, respectively, and the low-dimensional latent space $Z$. The dimensionality of the latent and observation spaces is denoted by $Q, D, L$, respectively, and $N$ denotes the number of objects / data points (quasars). Note that the correlation between the two observation spaces is not explicitly but implicitly modelled through a shared latent space. Generating a single data point $(\boldsymbol{x}_n, \boldsymbol{y}_n)$ (a row across $X$ and $Y$) entails a forward pass through the GPs, where $\boldsymbol{x}_n = [\ldots, x_{nd}, \ldots]$ is generated as $[f_1(\boldsymbol{z}_n), f_2(\boldsymbol{z}_n), \ldots, f_D(\boldsymbol{z}_n)]$ and $\boldsymbol{y}_n = [\ldots, y_{nl}, \ldots]$ is generated as $[f_{D+1}(\boldsymbol{z}_n), f_{D+2}(\boldsymbol{z}_n), \ldots, f_{D+L}(\boldsymbol{z}_n)]$.

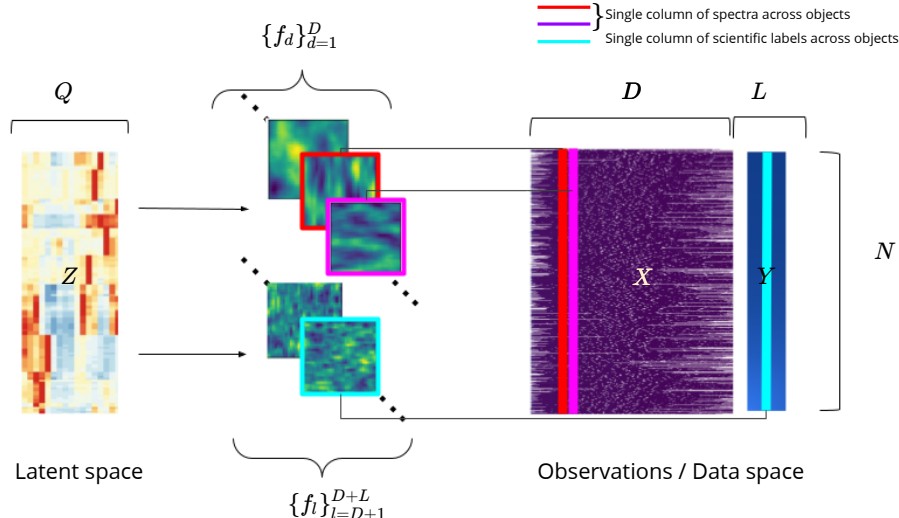

Figure 2: Shared GPLVM with multiple observation spaces. The blocks on the right-hand side denote the double observation spaces $(X, Y)$ of quasar spectra and scientific labels respectively. In the center are two stacks of GPs, one for each observation space which control the data generation process through the shared latent space. In the figure above we assume $Q = 2$ (for ease of visualisation) since we denote the GPs are two dimensional surfaces, however, typically $Q$ can be higher than 2 corresponding to higher dimensional GPs.

## 5 Algorithm

We enclose the pseudo-code in Algorithm 2 for stochastic variational inference in the context of the shared model for clarity. Let $\mathcal{L}_x$ and $\mathcal{L}_y$ denote the ELBO's for each of the observation spaces and let $\mathcal{L}_x^{(B)}$ and $\mathcal{L}_y^{(B)}$ denote the ELBOs formed with a randomly drawn mini-batch of the data (across all dimensions). For a mini-batch (subset) of the data $X_B \subset X$, the mini-batch ELBO is given by,

$$\mathcal{L}_x \simeq \mathcal{L}_x^{(B)} = \frac{N}{B} \left( \sum_{b,d} \langle \log p(x_{b,d} | \boldsymbol{f}_d, \boldsymbol{z}_b, \sigma_x^2) \rangle_{p(\boldsymbol{f}_d | \boldsymbol{u}_d, Z) q(\boldsymbol{u}_d)} + \sum_b \log p(\boldsymbol{z}_b) \right) - \sum_d \mathrm{KL}(q(\boldsymbol{u}_d) || p(\boldsymbol{u}_d | \tilde{Z})) \quad (19)$$

where the scaling term is important for the mini-batch ELBO to be an estimator of the full-dataset ELBO.

### 5.1 Computational Cost

The training cost of the canonical stochastic variational GPLVM is dominated by the number of inducing points $\mathcal{O}(M^3 D)$ (free of $N$) where $M \ll N$ and $D$ is the data-dimensionality (we have $D$ GP mappings $f_d$, one per output dimension). The practical algorithm is made further scalable with the use of mini-batched learning. In our shared model with two sets of GPs, the dynamics of the training cost are the same except that they increase linearly in the number of additional dimensions ($L$), making the cost $\mathcal{O}(M^3(D + L))$. The number of global variational parameters to be updated in each step (parameters of $q(U)$) is $MQ + M(D+L) + M^2(D+L)$, where $MQ$ are the $M$ $Q$-dimensional inducing inputs $\tilde{Z}$ (shared), $M(D+L)$ is the size of the mean parameters of the inducing variables $\boldsymbol{u}_d$, $\boldsymbol{u}_l$ and $M^2(D+L)$ are the full-rank covariances of the inducing variables. The local variational parameters $Z$ (the latent embedding shared across GPs) are of size $NQ$ and model hyperparameters (kernel hyperparameters) are of size $2Q + 4$, which account for $Q$ input lengthscales, a scalar signal variance and noise variance per GP group $\{f_d\}$ and $\{f_l\}$. We use the squared exponential kernel with automatic relevance determination across both sets of GPs.

## 6 Related methodological work

In this section we present related work in multi-output Gaussian processes in more detail. Canonical multi-output Gaussian processes almost always refer to the supervised framework or multi-output regression

---

**Algorithm 1:** Training Framework

---

**Input:** ELBO objective $\mathcal{L} = \mathcal{L}_x + \mathcal{L}_y$, gradient based optimiser `optim()`, observation spaces $X$ (spectra) and $Y$ (scientific labels)

Initial model params:

   $\theta = (\theta_x, \theta_y)$ (covariance hyperparameters for GP mappings $f_{1:D}, f_{D+1:D+L}$),

   $\sigma^2 = (\sigma_x^2, \sigma_y^2)$ (variance of the noise model for each likelihood),

   $Z \equiv \{z_n\}_{n=1}^N$ (point estimates for latent embedding)

Initial variational params:

   $\tilde{Z} \in \mathbb{R}^{M \times Q}$ (inducing locations),

   $\lambda = \{m_h, S_h\}_{h=1}^{D+L}$ (global variational params for inducing variables per dimension $\boldsymbol{u}_h$),

**while** *not converged* **do**

     • Choose a random mini-batch of the data from both the observation spaces $X_B \subset X, Y_B \subset Y$.

     • Form a mini-batch estimate of the ELBO: $\mathcal{L}_x^{(B)} + \mathcal{L}_y^{(B)}$

     • Gradient step for global parameters $\boldsymbol{g} \leftarrow \nabla_{\theta, \sigma^2, \tilde{Z}, \lambda}(\mathcal{L}_x^{(B)} + \mathcal{L}_y^{(B)})$

     • Gradient step for local parameters $\boldsymbol{l} \leftarrow \nabla_{Z_B}(\mathcal{L}_x^{(B)} + \mathcal{L}_y^{(B)})$ (where $Z_B$ are the latent embeddings corresponding to points in the mini-batch)

     • Update all parameters $\tilde{Z}, \theta, \sigma^2, \lambda, Z_B \equiv \{z_b\}_{n=1}^B \longleftarrow$ `optim()` using gradients $\boldsymbol{g}, \boldsymbol{l}$

**end**

**return** $\theta, \sigma^2, \tilde{Z}, \lambda, Z$

---

**Algorithm 2:** Prediction Framework

---

(Predict $Z^*$ corresponding to test $X^*, Y^*$)

**Input:** Trained global and local parameters $\theta, \sigma^2, \tilde{Z}, \lambda, Z$, test observation spaces $X^*$ (spectra) and $Y^*$ (scientific labels).

1. Initialise latent embedding $Z^* \equiv \{z_{n^*}\}_{n^*=1}^{N^*}$ corresponding to test points.

2. Extend the joint ELBO to include terms corresponding to the $N^*$ additional data points.

   $\mathcal{L}_x^* \leftarrow \mathcal{L}_x + \sum_{n^*, d} \langle \log p(x_{n^*, d} | \boldsymbol{f}_d, z_{n^*}, \sigma_x^2) \rangle_{p(\boldsymbol{f}_d | \boldsymbol{u}_d, Z) q(\boldsymbol{u}_d)} + \sum_{n^*} \log p(z_{n^*})$

   $\mathcal{L}_y^* \leftarrow \mathcal{L}_y + \sum_{n^*, l} \langle \log p(y_{n^*, l} | \boldsymbol{f}_l, z_{n^*}, \sigma_y^2) \rangle_{p(\boldsymbol{f}_l | \boldsymbol{u}_l, Z) q(\boldsymbol{u}_l)} + \sum_{n^*} \log p(z_{n^*})$

   $\mathcal{L}^* \leftarrow \mathcal{L}_x^* + \mathcal{L}_y^*$

3. Freeze all global and local parameters except for $Z^*$

**while** *not converged* **do**

     • Gradient step for $Z^*$: $\boldsymbol{l}^* \leftarrow \nabla_{Z^*} \mathcal{L}^*$

     • Update $Z^* \leftarrow$ `optim()` using gradients $\boldsymbol{l}^*$.

**end**

**return** $Z^*$

*(Note that the gradients of $\mathcal{L}_x$ and $\mathcal{L}_y$ with respect to $Z^*$ are 0 and the only terms that are optimised are the additional terms corresponding to the new data points.)*

---

(Álvarez et al., 2010) where the targets $\{\boldsymbol{x}_d(\boldsymbol{z})\}_{d=1}^D$ are multidimensional, continuous and Gaussian distributed corresponding to inputs $\boldsymbol{z} \in \mathbb{R}^p$. The main focus is on defining a suitable cross-covariance function between the outputs, this allows treatment of the outputs as a single GP with a suitable covariance function (Alvarez et al., 2012). The intrinsic and linear co-regionalization models (Goovaerts, 1993; Journel & Huijbregts, 1976) are a popular approach in which each output is modelled as a weighted sum of shared latent functions (GP) where typically the number of latent processes is smaller than the number of outputs enabling efficiencies. To some extent, multitask learning with GPs can be viewed as an instance of multi-output learning where we want to avoid tabula rasa learning for each task and evolve a framework for sharing information between multiple tasks (Bonilla et al., 2007). Further, there is the simplistic paradigm where each output is modelled with its own independent single-output GP and no correlation between the outputs is assumed. This approach, while easy to implement, is severely limited in its ability to jointly model the outputs.

In the unsupervised paradigm, the starting point is a high-dimensional data matrix $N \times D$. The conventional Gaussian process latent variable model (GPLVM) operates like a multi-output model by default where each column of the data is modelled by an independent GP on the same shared set of inputs, kernel function

| Metrics ($\longrightarrow$) | Mean Absolute Error | | |
|---|---|---|---|
| Models ($\longrightarrow$) | Baseline | Shared (ours) | Specialised (ours) |
| Attributes ($\downarrow$) | | | |
| Spectra | $0.1103 \pm 0.0035$ | $0.0972 \pm 0.0024$ | $0.0969 \pm 0.0013$ |
| Bolometric Luminosity | $0.2444 \pm 0.0043$ | $0.2144 \pm 0.0033$ | $\mathbf{0.2069 \pm 0.0014}$ |
| Black hole mass | $0.2712 \pm 0.0031$ | $\mathbf{0.2319 \pm 0.0028}$ | $0.2362 \pm 0.0025$ |
| Eddington Ratio | $0.2525 \pm 0.0070$ | $\mathbf{0.2118 \pm 0.006}$ | $0.2226 \pm 0.0015$ |

Table 1: Summary of test-time reconstruction abilities. Mean absolute error on denormalised data ($\pm$ standard error of mean) evaluated on average of 5 splits with 75% of the data used for training in the 22k dataset. The shared and specialised versions of the model outperform the baseline model in all the reconstruction tasks. In the shared vs. specialised comparison the performance difference is not statistically significant for the spectra reconstruction but overall favours the shared model with hyperparameter sharing across the scientific label reconstruction.

and hyperparameters. The GPLVM is a decoder-only model, and some of their prominent variants are the back-constrained GPLVM (Lawrence & Quiñonero Candela, 2006), the supervised GPLVM (Jiang et al., 2012), the discriminative GPLVM (Urtasun & Darrell, 2007) and the shared GPLVM (Ek, 2009). The latter considers the task of dealing with multiple views or observation spaces (each of which is high-dimensional). This work is built on the idea of a shared latent space underlying the multiple observation spaces similar to Ek et al. (2007) but adapts it for scalable inference using stochastic variational inference (SVI). The stochastic variational GPLVM was proposed in Lalchand et al. (2022), but only considered a single observation space as in a canonical GPLVM.

## 7 Experiments

In this section, we demonstrate a range of experiments aimed at assessing the reconstruction quality of test (unseen) quasar spectra and scientific attributes, as well as the robustness of uncertainty quantification by computing the negative log predictive density (NLPD) of test labels $-\log p(Y^*|Z^*)$ under the predictive distribution where $Z^*$ has been informed by both modalities (spectra and labels) and only spectra.

The data used in this work are quasar spectra observed as part of the Sloan Digital Sky Survey (SDSS) DR16 (Lyke et al., 2020). We chose all quasars with spectra that have a signal-to-noise ratio (SNR) per pixel $> 10$, which results in a total of 22844 quasar spectra. The observed spectra are shifted into the rest frame wavelength space, re-binned onto a common wavelength grid, and flux normalized to unity at around 2500 Å. We mask strong absorption lines that might arise in the spectra due to foreground galaxies along our line-of-sight to the quasar, and are thus not intrinsic spectral features of the quasar. The four scientific labels for these quasars are (1) their SMBH mass, (2) their bolometric luminosity, i.e. the total power output across all electromagnetic wavelengths, (3) their redshift which denotes the factor by which the emitted wavelengths have been "stretched" due to the expansion of the universe, and (4) their Eddington ratio, which is a measure of the accretion and growth rate of the SMBH. All measurements were previously uniformly determined by Wu & Shen (2022). We conduct experiments across two data sets with 1k (see Table 4) and 22k points (as an ablation to assess performance with a smaller dataset). The "Baseline" model in the experiments refers to the canonical stochastic variational GPLVM (Lalchand et al., 2022) which treats multiple observation spaces using the same set of independent GPs learning a single set of kernel hyperparameters. The "Specialised" model refers to the shared model but with individual hyperparameters for each scientific label.

### 7.1 Reconstructing Quasar spectra

We assess the quality of our probabilistic generative model in reconstructing test quasar spectra. At test time we deal with spectra and scientific labels from test quasars denoted by $(X_{\text{gt}}^*, Y_{\text{gt}}^*)$ (we use the index 'gt' to denote ground truth). The 2-step prediction learns the low-dimensional shared latent variables $Z^*$ (point estimate per data point) which acts as input to the GP decoder predicting the corresponding spectra $Z^* \rightarrow X_{\text{est}}^*$. Note that the ground truth spectra contains several missing pixels (dimensions) and

the probabilistic decoder provides a reasonable reconstruction at those locations. In Fig. 3 we visualise the reconstruction (posterior predictive mean) of four test quasars along with ground truth measurements and 95% prediction intervals. We achieve remarkably good reconstruction estimates from the latents; further, the prediction intervals capture the ground truth spectra providing robust coverage at peaks and extrapolated regions.

### 7.1.1 Reconstructing missing spectra

In this experiment, we test the generative model's ability to learn from massively missing chunks of the spectra at test-time. We observe a partial window of the spectra in each plot (given by the shaded region in Fig. 4), hence the latent variables corresponding to these points are only informed by the observed region. We then reconstruct the whole spectrum from the latent variables informed by the partial spectra. We enclose our results in Fig. 4. Reconstruction entails the following inference steps: $X^*_{\text{partial}} \to Z^* \to X^*_{\text{full}}$. We note that the quality of the mean prediction deteriorates compared to the fully observed test point predictions. However, the coverage of the prediction intervals is robust even as we move away from the shaded observed regions. Furthermore, if the model is given a spectral region with very little information (e.g., in the bottom panel of Fig. 4, the shaded region contains information only about the quasar's continuum), the uncertainties increase significantly, as one would expect.

## 7.2 Predicting scientific labels *only* from spectra $X^*$

The dimensions $L$ corresponding to the scientific labels in the dataset governed by their own GP decoders $\{f_l\}_{l=D+1}^{D+L}$ are a critical prediction quantity. The ability to reconstruct these quantities from learned latent variables is an important test of the generalizability of our model. Very often astronomers want to reason about the scientific attributes of quasars just by analysing their spectra. In this experiment, we demonstrate precisely this use case in which latent variables $Z^*$ are informed only by spectra $X^*$, computing the cross-modal prediction involves learning $Z^*$ from the ground truth spectra and then using just $Z^*$ to predict the scientific labels $Y^*$, succinctly, we can write these steps as: $X^* \to Z^* \to Y^*$.

In Fig. 5 we demonstrate the accuracy of our reconstructions by plotting each of the dimensions against ground truth held-out data. We show reconstructions for 200 test points randomly sampled from the full test set. Each point in the scatter denotes a quasar, and the x-axis denotes the ground truth measurement. The orange vertical error bars denote the intervals $1\sigma$ computed by extracting the diagonals from the predictive posterior GP for each dimension. We can observe a high degree of prediction accuracy across the three scientific labels and, furthermore, the reconstruction quality is weakly correlated to the spectral signal-to-noise ratio (SNR) (at least beyond the data quality cut of SNR > 10 which we apply as a preprocessing step); this is mainly due to acute data imbalances where more than 90% of the objects appear at the lower SNRs affecting prediction quality and generalization for the quasars at higher SNRs (see Appendix D). Note that we did not attempt to reconstruct the redshift label from the spectra $X$. This is because the spectra have already been normalized to account for their redshift. Specifically, the spectra were shifted to rest-frame wavelength space by dividing the observed wavelengths by $1 + z$ (where $z$ is the redshift). As a result, normalized spectral shapes no longer contain significant information about their redshift (as shown in Yang et al. (2021) and Onorato et al. (2024)).

However, we still include the redshift information in our multimodal GPLVM. This is because excluding it marginally weakens the results, both for spectra reconstruction and label prediction. We hypothesize that, while the spectral shapes themselves do not carry redshift information, the patterns of missing pixels may still encode some redshift-related information.

From Table 2 we can see that the variability of NLPD estimates is much higher under the cross-modal protocol of decoding scientific labels on the basis of the spectrum. In other words, reconstructing scientific labels from spectra alone has a much higher uncertainty than reconstructing scientific labels from both modalities. As a result, the shared model offers only a subtle improvement over the baseline model in terms of uncertainty quantification. In the fully observed protocol, the shared model is more confident (Table 2) and accurate (Table 1) in its predictions relative to the baseline model. It is also important to note that under the much harder cross-modal protocol, the performance of the shared model surpasses that of the baseline model.

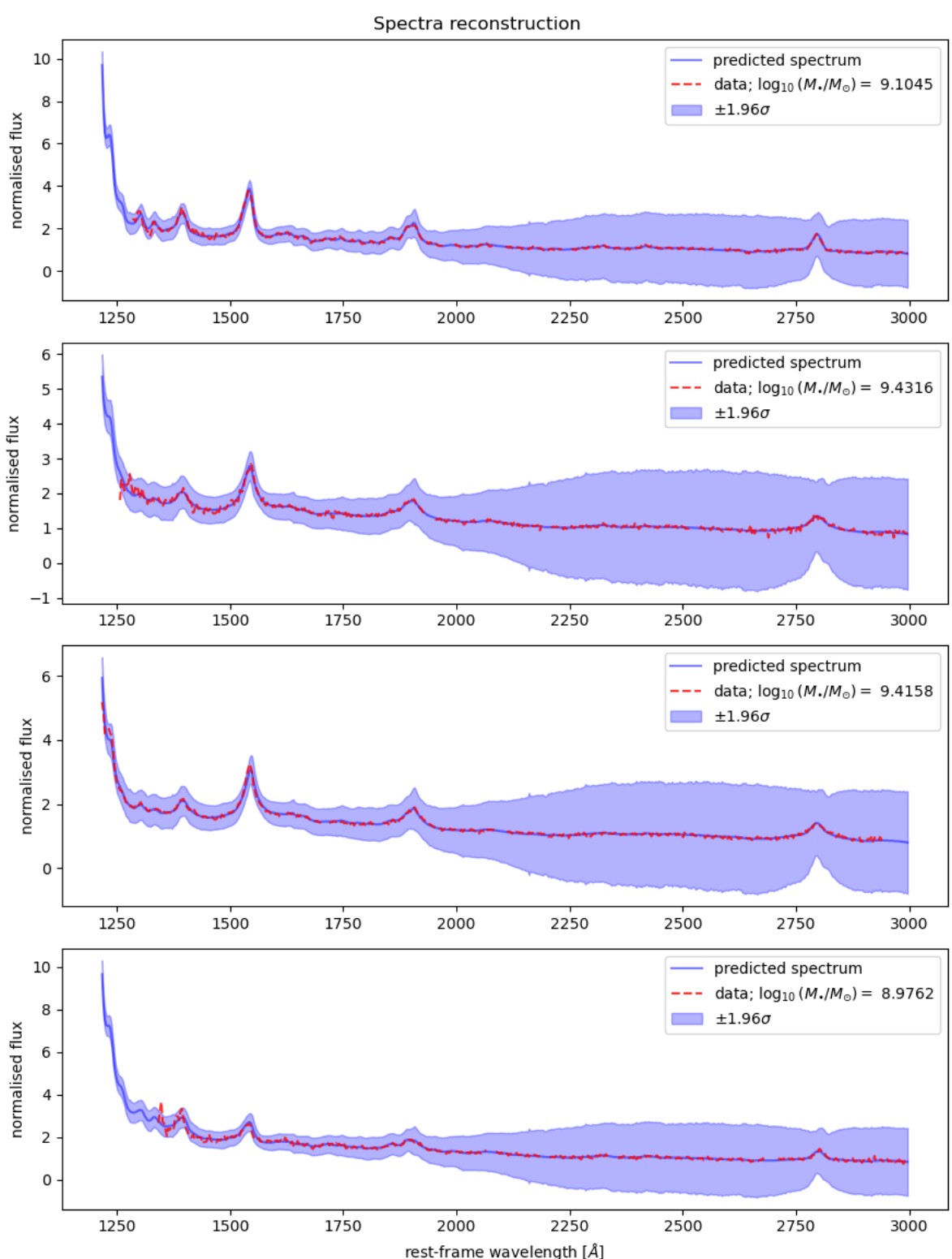

Figure 3: Reconstruction plots of test quasar spectra with $\pm 1.96\sigma$ intervals. The blue curve denotes the posterior predictive mean at each dimension.

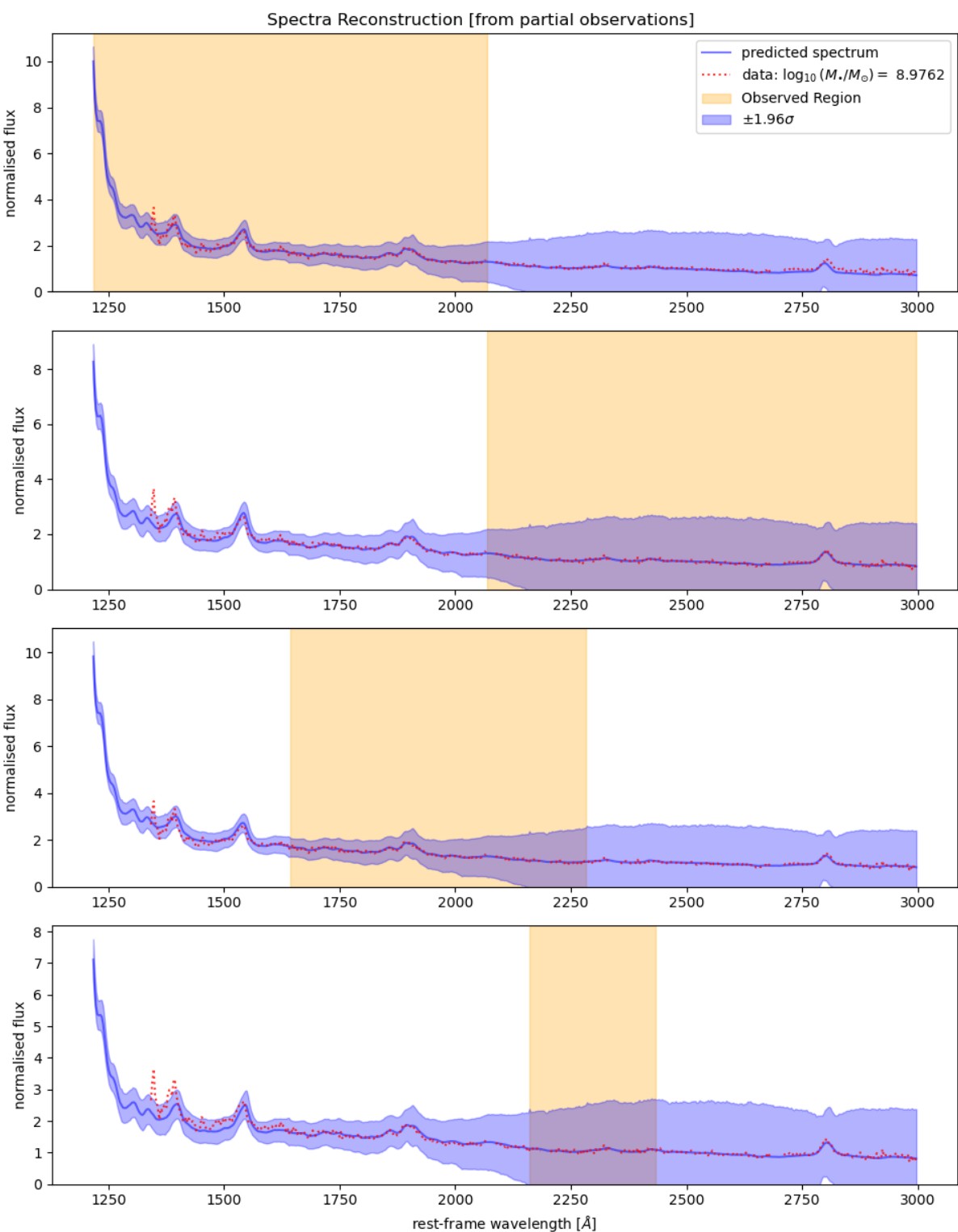

Figure 4: Reconstruction of a single spectra from the latent informed by a partially observed spectrum. The shaded orange regions denote the "observed" wavelength regions for this experiment. Note in the 4th panel the 95% prediction intervals are wider as they were observed over a shorter and less informative wavelength window.

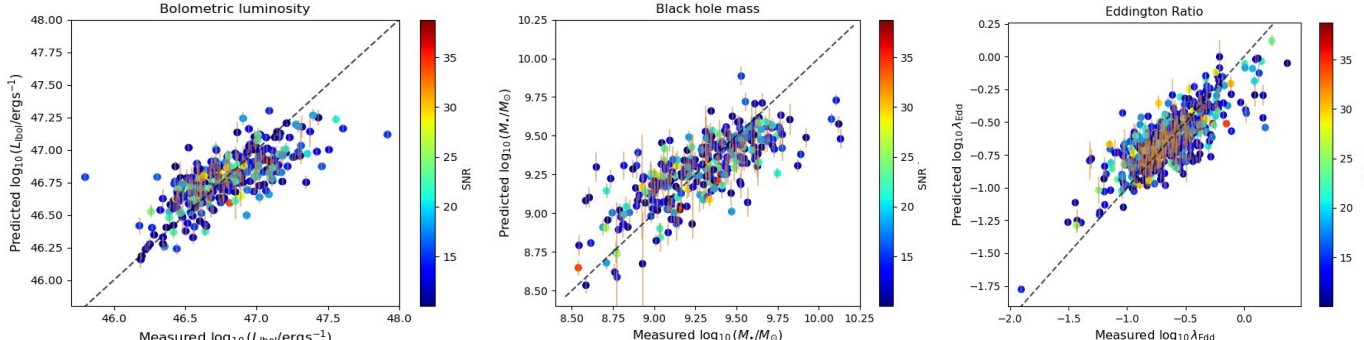

Figure 5: Scientific label prediction for the quasars' bolometric luminosity (left), black hole mass (middle) and Eddington ratio (right) colored by the SNR of their spectra based on test $X^*$ **only**. The dashed black line (- - -) denotes the 1-to-1 line to aid visualisation of reconstruction accuracy. The vertical and horizontal errorbars (——) denotes posterior predictive standard deviation.

| Experiment | Luminosity | Black hole mass | Eddington Ratio |
|---|---|---|---|
| Baseline (Fully observed) | $0.076 \pm 0.0012$ | $0.148 \pm 0.0043$ | $0.1178 \pm 0.0031$ |
| Baseline (Spectra observed) | $0.1143 \pm 0.042$ | $0.1742 \pm 0.0621$ | $0.1376 \pm 0.0646$ |
| Shared (Fully observed) | **$-0.029 \pm 0.0018$** | **$0.1009 \pm 0.0013$** | **$0.0702 \pm 0.018$** |
| Shared (Spectra observed) | $0.0862 \pm 0.051$ | $0.1604 \pm 0.0245$ | $0.1294 \pm 0.0461$ |

Table 2: Summary of test-time uncertainty quantification under the full and partial reconstruction framework. Mean negative log predictive density (lower is better) on de-normalised data across 5 train/test splits. A higher NLPD indicates lower confidence in the predictions and the model indicates this when reconstructing the labels of a quasar just on the basis of its spectrum. The fully observed model corresponds to predictions based on test data when the all modalities are observed and reconstructed, $(X_{\text{gt}}^*, Y_{\text{gt}}^*) \to Z^* \to Y_{\text{est}}^*$ or when only the spectrum of the quasar is informing the latent, $(X_{\text{gt}}^* \to Z^* \to Y_{\text{est}}^*)$. The latents are in turn used to decode the scientific labels.

### 7.3 Generating spectra corresponding to synthesized labels: an ablation study

In this experiment, we demonstrate the ability of the model to generate spectra corresponding to synthesized scientific labels. Concretely, we simulate artificial labels by systematically varying only one of the labels within a reasonable range in each plot in Fig. 6. The range of variation for each label is summarized by the colorbar in each plot, for instance, in the black hole mass simulation we generate 100 spectra ($X^*$) corresponding to simulated labels ($Y^*$) of black hole masses $\log_{10}(M_\bullet/M_\odot)$ in the range $[7.9, 10.0]$; in order to ablate the influence of other labels (redshift, bolometric luminosity, and the Eddington ratio) on the spectra we keep their values fixed to mean values computed from the training dataset. Lastly, we can summarize the inference steps as the inverse of the previous section: $Y^* \to Z^* \to X^*$. The ability for prediction and generation of cross-modalities is an important strength of our design.

Reassuringly, the model exhibits the expected behaviour. For instance, the emission lines are broader for quasars with higher black hole masses, and the spectral dependence with bolometric luminosity shows e.g. in the Mg, II line at $\approx 2800$Å the well-known Baldwin effect (Baldwin, 1977), which indicates that quasar spectra show a decreasing equivalent width of their UV and optical emission lines with increasing bolometric luminosity.

In addition, the gray bars denote the 95% prediction intervals averaged across the 100 spectra at each dimension. The uncertainty intervals are wider at higher wavelengths, as there is a high concentration of missing pixels in the training data at those wavelengths; hence, there is greater uncertainty about the generated spectra.

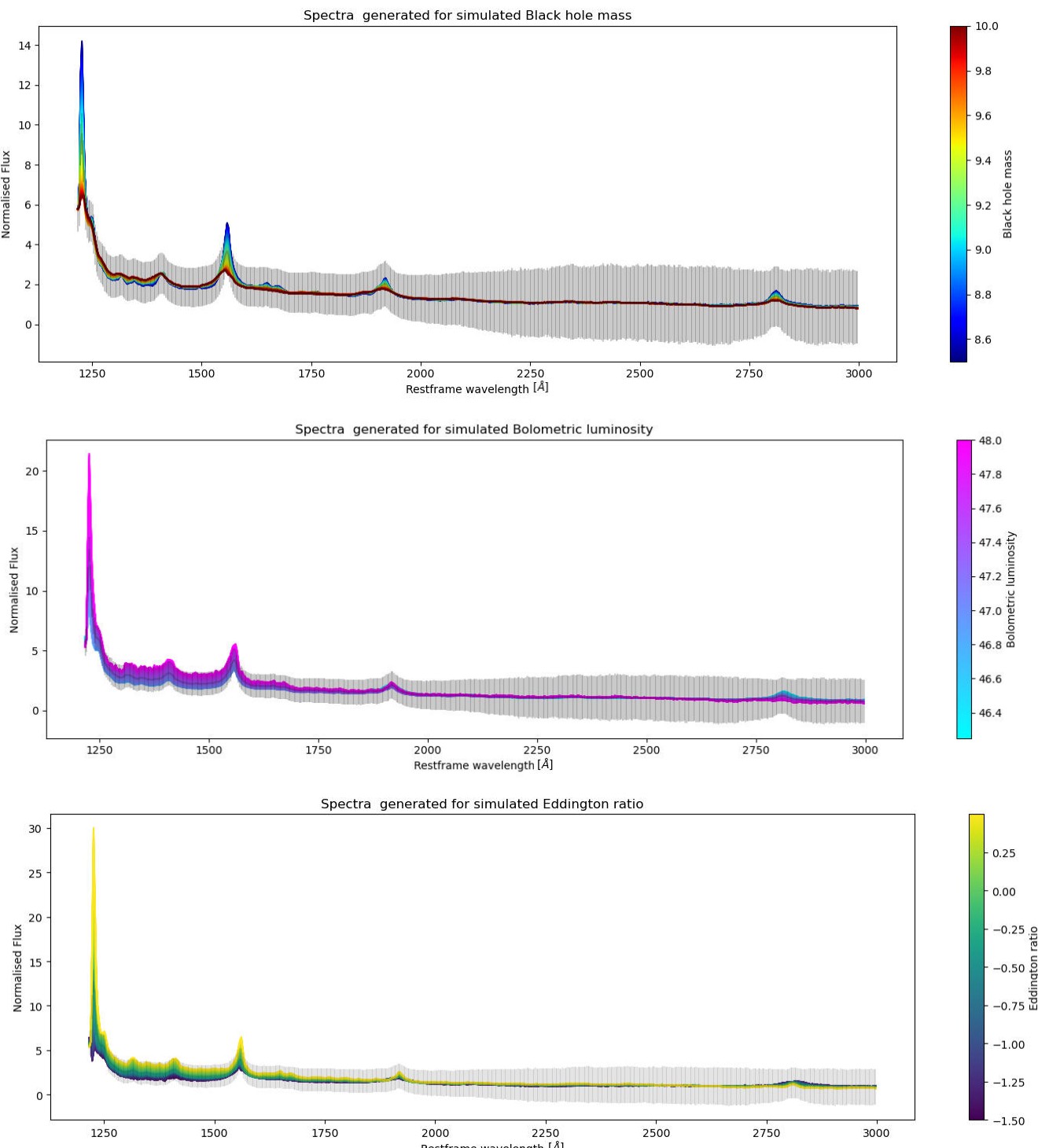

Figure 6: An experiment demonstrating cross-modal prediction $Y^* \to Z^* \to X^*$. In the plots we show generated quasar spectra for simulated labels for black hole mass, bolometric luminosity and Eddington ratio. In each plot we vary the respective scientific label in a reasonable range (shown by the range on the colorbar) while keeping the other labels to fixed values.

### 7.4 On benchmarking with a traditional regression approach

The dependency between the spectra and labels is modelled through the shared latent space, which generates both views of the data. Without a generative model it would not be possible to generate spectra corresponding to scientific labels (generate $X$ corresponding to $Y$), we would only be able to predict labels based on fixed observed spectra. With a joint generative model, it is possible to generate both spectra and labels or generate one conditioned only on observing the other view through a jointly learned compression. In other words, we can extract the shared latent ($Z$) corresponding to $X$ (spectra) to predict a reconstructed estimate of the spectra and the labels $(\hat{X}, \hat{Y})$, or the other way round $Y \xrightarrow{optimise} Z \xrightarrow{decode} (\hat{X}, \hat{Y})$ as shown in experiments. Further, the spectra are only partially observed with several missing pixels making canonical regression less straightforward as one would need to account for missing/uncertain inputs. In the case of training a GP to predict labels directly from fixed high-dimensional spectra, one would have to impute the missing pixels at the outset, as the prior covariance matrix cannot be computed on missing inputs.

## 8 Limitations

The primary limitation of the model arises from the fact that GPLVMs are decoder-only models, they constitute a (potentially) smooth mapping from the latent space to the data space. This means that points close in latent space will be close in data space but not the other way round. Data space similarities can be preserved by including a back-constraint or an encoder that additionally maps the data to latent space (Lawrence & Quiñonero Candela, 2006; Bui & Turner, 2015). The encoder usually takes the form of a neural network, but other choices are possible. As the title of the manuscript emphasizes, the model we propose is a decoder-only model. The absence of an encoder complicates the test-time inference as there is no deterministic way to access the latent points $\boldsymbol{z}^*$ corresponding to the new unseen observation $(\boldsymbol{x}^*, \boldsymbol{y}^*)$. Inferring the latent point corresponding to the unseen test point entails freezing the model and variational parameters post-training and re-optimizing the ELBO objective subject to $(\boldsymbol{z}^*)$ with the new data point(s) included (see Algorithm 2).

This optimization procedure for test-time inference ends up being too inconvenient in contrast to encoder-decoder models such as VAEs (Kingma & Welling, 2013), where inferring a latent point corresponding to an unseen data point entails a forward pass through the trained encoder network (constant-time predictions $\mathcal{O}(1)$). The main reason an extension to an autoencoded shared stochastic GPLVM is not straightforward is due to the presence of missing data. The encoder network must be capable of handling arbitrarily missing dimensions. Methods like the partial VAE (Ma et al., 2018a;b) address this challenge in the context of VAEs where each observation is augmented with a column index indicating the observed dimension; the encoder network then processes these tuples as a 'set' with a permutation invariant set encoding function. A similar approach can be straightforwardly adapted from our shared latent space setup with Gaussian process decoders. The combination of a set encoder (to process missing dimensions) and a non-parametric decoder has not appeared in the literature to the best of our knowledge. We are currently working on incorporating this feature into our framework. Note that the presence of an encoder acts as a constraint in the model, and there is an inherent trade-off in terms of faster test inference and marginally weaker reconstructions / predictions.

## 9 Methodological extensions for future work

A natural extension of the shared GPLVM proposed here is to incorporate an encoder, bringing the whole architecture more in line with modern deep-generative autoencoder models while simultaneously preserving the advantage of uncertainty quantification in the outputs. Bui & Turner (2015) propose the canonical GPLVM with an encoder and train it with SVI, therefore, a natural extension is to demonstrate it for the shared setting. The shared setting in the data context presented here opens up questions about the encoding of partially observed vectors along with fully observed ones, and masked encoders (He et al., 2022) might be a relevant architectural paradigm to explore.

Another extension is related to the structure of the kernels, since the kernels factorise over dimensions and are closed over the multiplicative operation, one approach is to select a composite product kernel over dimensions of the latent space instead of two sets of GPs with their own respective kernels. The former approach (not studied here) induces a partitioning of the latent space into dimensions which would cater to specific views of the data and can be modelled with individual kernel functions. In our context, we did not see the gain in the mixture kernel formulation; note that this approach raises an additional question as to how many latent dimensions to allocate to each kernel function. Nevertheless, partitioning the latent space and modelling with mixed or composite kernels could be beneficial in specific settings. More experiments are required to understand the exact settings in which this approach may outshine the alternative framework presented here.

## 10  Scientific Interpretation and Significance

Our new generative model allows us to *simultaneously* model the spectral properties of quasars as well as their scientific labels, thus opening up novel possibilities to study the evolution of quasars and their dependency on physical parameters governing the SMBH growth. In the following, we will highlight just two possible and exciting future applications of this work.

Astronomers observe SMBHs with billions of solar masses in size in the center of very distant, high-redshift quasars, at a time when the universe is still in its infancy and only a few hundred Myr (million years) old. This rapid growth of SMBHs in the very short amounts of available cosmic time has been an open puzzle for decades, and it has been argued that very high accretion rates in excess of the theoretical upper limit, the so-called Eddington limit, are required to explain the rapid black hole growth. However, obtaining precise black hole mass measurements of quasars is challenging and time-intensive as it requires multi-epoch observations of certain emission lines in the quasar spectra (Peterson, 1993; Barth et al., 2015). This procedure becomes increasingly challenging for very distant, high-redshift quasars, as relativistic time-dilation effects require longer timespans of these observations. Furthermore, the traditionally used rest-frame optical emission lines to calibrate the black hole masses are unobservable with ground-based observatories, as these optical wavelengths have been shifted to the infrared at larger distances, which require space-based telescopes, such as NASA's recently launched James Webb Space Telescope. Using the new generative model provides us with the possibility to determine the masses of SMBHs for quasars at all redshifts from single-epoch data observed with ground-based observatories, as it does not require wavelength coverage of specific emission lines; further, it can handle missing data.

Furthermore, we have shown that our model is able to predict other physical properties of quasars such as their bolometric luminosities (see Fig. 5). This suggests that we can obtain a measurement of the quasars' absolute luminosities using their spectral information alone, which provides a new opportunity to use quasars as the so-called "standard candles." Standard candles are incredibly valuable for astronomy, as knowing the luminosity of an object allows one to determine its distance. Famously, supernovae have been used as standard candles, which led to the Nobel Prize winning discovery of the expansion of our universe and the existence of dark energy (Riess et al., 1998). The use of quasars as standard candles has previously been suggested leveraging the relationship between the X-ray and UV luminosities of quasars to determine their distances (e.g. Lusso & Risaliti, 2017; Sacchi et al., 2022). Our generative model enables predictions of the quasars' bolometric luminosities leveraging these spectral dependencies on luminosity, enabling the use of quasars as standard candles (or "standardizable candles") to map the expansion of the universe to larger distances than possible with supernovae. Further testing of the required precision with which the quasars' luminosities and thus distances can be predicted by the generative model will be necessary in order to determine whether the constraints on the universe's expansion rate obtained by the more numerous quasars can improve on the current constraints by the fewer but very accurately measured distances from supernovae.

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

## A  The evidence lower bound $\mathcal{L}_x$

In this section, we explicitly show the derivation of expected log-likelihood term in Eq. (5):

$$p(X) \geq \int p(F|U,Z)q(U) \log p(X|F,Z)dFdU - \mathrm{KL}(q(U)||p(U)) + \log p(Z) \tag{20}$$

$$
\int p(F|U,Z)q(U) \log p(X|F,Z)dFdU = \int q(\boldsymbol{u}_d) \int p(\boldsymbol{f}_d|\boldsymbol{u}_d, Z) \log \prod_{n=1}^{N}\prod_{d=1}^{D} \mathcal{N}(x_{n,d}; f_d(\boldsymbol{z}_n), \sigma_x^2)d\boldsymbol{f}_d\boldsymbol{u}_d
$$
$$
= \int q(\boldsymbol{u}_d) \int p(\boldsymbol{f}_d|\boldsymbol{u}_d, Z) \sum_{n,d} \log \mathcal{N}(x_{n,d}; f_d(\boldsymbol{z}_n), \sigma_x^2)d\boldsymbol{f}_d\boldsymbol{u}_d
$$
$$
= \sum_{n,d} \langle \log p(x_{n,d}|\boldsymbol{f}_d, \boldsymbol{z}_n, \sigma_x^2) \rangle_{p(\boldsymbol{f}_d|\boldsymbol{u}_d, Z)q(\boldsymbol{u}_d)} \tag{21}
$$

### A.1  Tractability of the expectation term

The expectation term above is not just factorisable but also tractable, we show this explicitly below:

$$
\int q(\boldsymbol{u}_d) \int p(\boldsymbol{f}_d|\boldsymbol{u}_d, Z) \sum_{n,d} \log \mathcal{N}(x_{n,d}; f_d(\boldsymbol{z}_n), \sigma_x^2)d\boldsymbol{f}_d\boldsymbol{u}_d \tag{22}
$$
$$
= \int q(\boldsymbol{u}_d) \sum_{n,d} \log \mathcal{N}(x_{n,d}; k_n^T K_{mm}^{-1}\boldsymbol{u}_d, \sigma_x^2) - \frac{1}{2\sigma_x^2}q_{n,n}
$$

$x_{n,d}$ is a scalar ($d^{th}$ dimension of point $\boldsymbol{x}_n$), $k_n^T$ is a matrix $1 \times M$ - the $n^{th}$ row of $K_{nm}$, we know that $p(\boldsymbol{f}_d|\boldsymbol{u}_d, Z) = \mathcal{N}(K_{nm}K_{mm}^{-1}\boldsymbol{u}_d, K_{nn} - K_{nm}K_{mm}^{-1}K_{mn})$. Further, $f_d(\boldsymbol{z}_n)$ is a scalar, denoting the value at index $\boldsymbol{z}_n$ of the vector $\boldsymbol{f}_d$. $q_{n,n}$ is the $n^{th}$ entry in the diagonal of the matrix $Q_{nn} = K_{nn} - K_{nm}K_{mm}^{-1}K_{mn}$. Next, performing the integration with respect to $q(\boldsymbol{u}_d) = \mathcal{N}(\boldsymbol{u}_d; \boldsymbol{m}_d, S_d)$ yields,

$$
\int q(\boldsymbol{u}_d)\Big[\sum_{n,d} \log \mathcal{N}(x_{n,d}; k_n^T K_{mm}^{-1}\boldsymbol{u}_d, \sigma_x^2) - \frac{1}{2\sigma_x^2}q_{n,n}\Big]d\boldsymbol{u}_d = \sum_{n,d}\Big[\log \mathcal{N}(x_{n,d}; k_n^T K_{mm}^{-1}\boldsymbol{m}_d, \sigma_x^2) - \frac{1}{2\sigma_x^2}q_{n,n}
$$
$$
- \frac{1}{2\sigma_x^2}\mathrm{Tr}(S_d\Lambda_n)\Big] \tag{23}
$$

Bringing it all together, the final lower bound can be written out explicitly as,

$$
\mathcal{L}_x = \sum_{n,d} \langle \log p(x_{n,d}|\boldsymbol{f}_d, \boldsymbol{z}_n, \sigma_x^2) \rangle_{p(\boldsymbol{f}_d|\boldsymbol{u}_d, Z)q(\boldsymbol{u}_d)} - \mathrm{KL}(q(U)||p(U)) + \log p(Z)
$$
$$
= \sum_{n,d}\Big[\log \mathcal{N}(x_{n,d}; k_n^T K_{mm}^{-1}\boldsymbol{m}_d, \sigma_x^2) - \frac{1}{2\sigma_x^2}q_{n,n} - \frac{1}{2\sigma_x^2}\mathrm{Tr}(S_d\Lambda_n)\Big] - \sum_d \mathrm{KL}(q(\boldsymbol{u}_d)||p(\boldsymbol{u}_d))
$$
$$
+ \sum_n \log p(\boldsymbol{z}_n) \tag{24}
$$

where all the data dependent terms factorise enabling mini-batching of gradients. The shared model uses a sum of ELBOs $\mathcal{L}_x + \mathcal{L}_y$ with a shared latent embedding $Z$, which we call the joint evidence lower bound (Eq. (15) in the main paper). The additive structure ensures that the joint ELBO is also factorisable across data points $N$.

```
+----------------------------------------------------------------------------+-----------+
|                                   Modules                                   | Parameters |
+----------------------------------------------------------------------------+-----------+
|                                inducing_inputs                             |   2500    |
|                                    Z.Z                                      |  208440   |
|       model_spectra.variational_strategy._variational_distribution.variational_mean |  164250   |
| model_spectra.variational_strategy._variational_distribution.chol_variational_covar |  41062500 |
|                        model_spectra.mean_module.constant                   |    657    |
|                   model_spectra.covar_module.raw_outputscale                |     1     |
|           model_spectra.covar_module.base_kernel.raw_lengthscale            |    10     |
|     model_labels.variational_strategy._variational_distribution.variational_mean |   1000    |
|  model_labels.variational_strategy._variational_distribution.chol_variational_covar |  250000   |
|                        model_labels.mean_module.constant                    |     4     |
|                   model_labels.covar_module.raw_outputscale                 |     1     |
|           model_labels.covar_module.base_kernel.raw_lengthscale             |    10     |
+----------------------------------------------------------------------------+-----------+
Total Trainable Params: 41689373
```

Figure 7: Number of trainable parameters for the 22k model where `model_spectra` refers to the GPs corresponding to $X$ and `model_labels` refers to the GPs corresponding to $Y$ observation space.

## B  Experimental set-up

In this section we detail the configuration of the experiments in Section 7 of the main paper. For each of the datasets, we repeat every experiment with five random seeds yielding different splits of the training data. The attributes of the data and sparse GP set-up are given in Table 3. We used a learning rate of 0.001 across

| Dataset | $N$ | $D$ | $L$ | Num inducing $M$ | Latent dim. $Q$ |
|---------|-----|-----|-----|------------------|-----------------|
| 1K      | 996 | 590 | 4   | 120              | 10              |
| 22K     | 22844 | 657 | 4 | 250              | 10              |

Table 3: Experimental configuration to reproduce experiments in section 3 of the main paper.

all parameters and ran the mini-batch loop with a batch size of 128 for 15,000 iterations on an Intel Core i7 processor with a GeForce RTX 3070 GPU with 8GB RAM memory. In order to give an estimate of the scale of the model for the 22k dataset we enclose a summary snapshot of the number of trainable parameters in our shared model.

## C  Testing on smaller sampled dataset

| Metrics ($\longrightarrow$) | Mean Absolute Error | |
|---|---|---|
| Models ($\longrightarrow$) | Baseline | Shared (ours) |
| Attributes ($\downarrow$) | | |
| Spectra | $0.0731 \pm 0.0020$ | $0.0707 \pm 0.0033$ |
| Black hole mass | $0.1882 \pm 0.0046$ | $\mathbf{0.1765 \pm 0.0054}$ |
| B. Luminosity | $0.1731 \pm 0.0043$ | $0.1658 \pm 0.0084$ |
| Eddington Ratio | $0.1707 \pm 0.0024$ | $\mathbf{0.1653 \pm 0.0026}$ |

Table 4: Summary of test-time reconstruction abilities for a smaller 1k dataset sampled with the same SNR data quality cut of SNR > 10. Mean absolute error on denormalised data ($\pm$ standard error of mean) evaluated on average of 5 splits with 90% of the data used for training in the 1k dataset. The performance improvement is not statistically significant for the spectra reconstruction and the bolometric luminosity prediction but favours the shared model for black hole mass and Eddington ratio.

## D   Sensitivity to SNR

In Fig. 8 we essentially plot the SNR per data point (test quasars) vs. the absolute error. The triangular scatter denotes a density imbalance: more data points are clustered at lower SNR values ( 10–15) compared to higher SNR values (>20). This is just an artefact of the data quality cut we deploy. The points seem evenly scattered across the range of absolute errors and SNR values. There doesn't appear to be a consistent trend where absolute error systematically increases or decreases with SNR. There is a very weak negative correlation (more discernable in the zoomed in plots in the bottom row) for the black hole mass and weak positive correlation for the luminosity. The unusual weak positive correlation can be better understood in the context of the count and missing pixels analysis (Fig. 9) in each SNR bin. There are very few high SNR (> 40) quasars in the dataset and further, these quasars also have more missing pixels than quasars at lower SNRs. Since the training dataset is biased towards lower SNR quasars, the model generalises poorly to higher SNR examples, for luminosity and Eddington ratio. It is highly likely that the pattern of missing pixels is similar among brighter objects and in turn among less luminous objects, making it difficult for the model to adapt to this shift at test time due to the acute data imbalance.

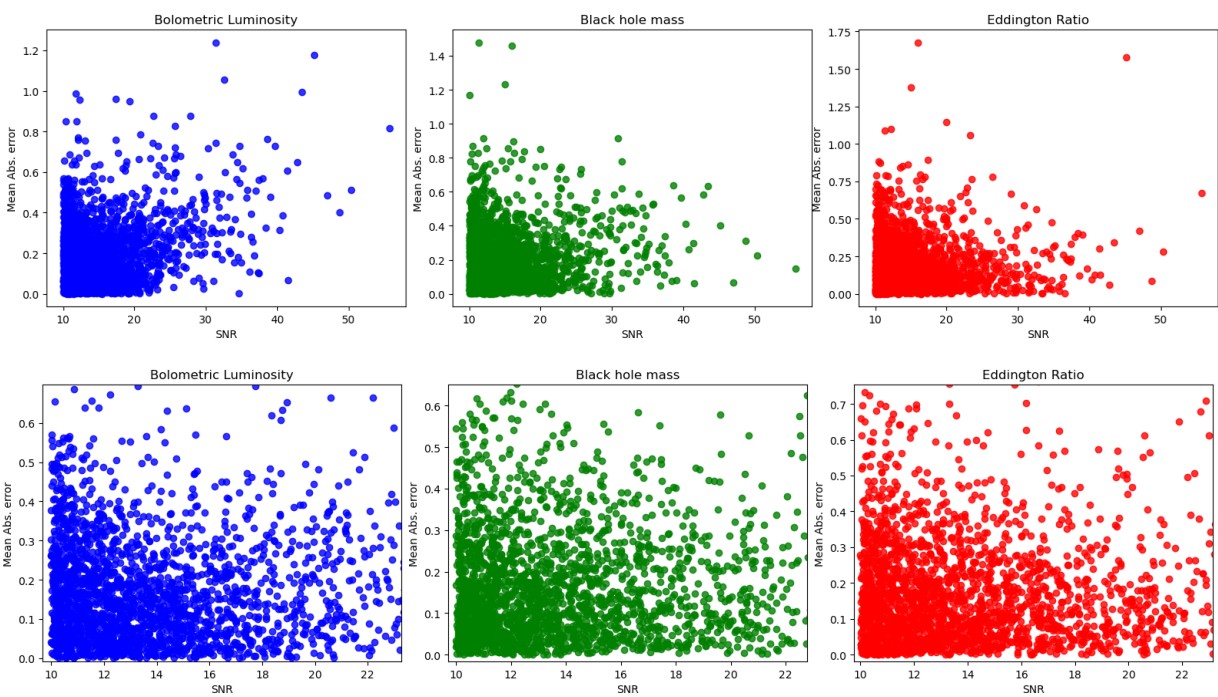

Figure 8: Top: SNR vs. error for test quasars, Bottom: SNR vs. error zoomed in on higher density regions

## E   Error vs. Uncertainty Calibration

In Fig. 10, we quantify uncertainty calibration by computing the mean predictive uncertainty for increasing error levels. We bin test points into five evenly spaced intervals over the range of the mean absolute error for each label. The x-axis denotes the centers of the bins for each label and the predictive uncertainty is computed as the square root of the variance — the diagonal of the GP posterior predictive distribution. In each label, we observe the trend of predictive uncertainty increasing with absolute error, although this effect is very subdued at lower error levels. Further, it is important to note that there is inherent noise in this calculation emanating from the differing number of points in each bin; this might explain some of the variability in the bolometric luminosity uncertainty estimates.

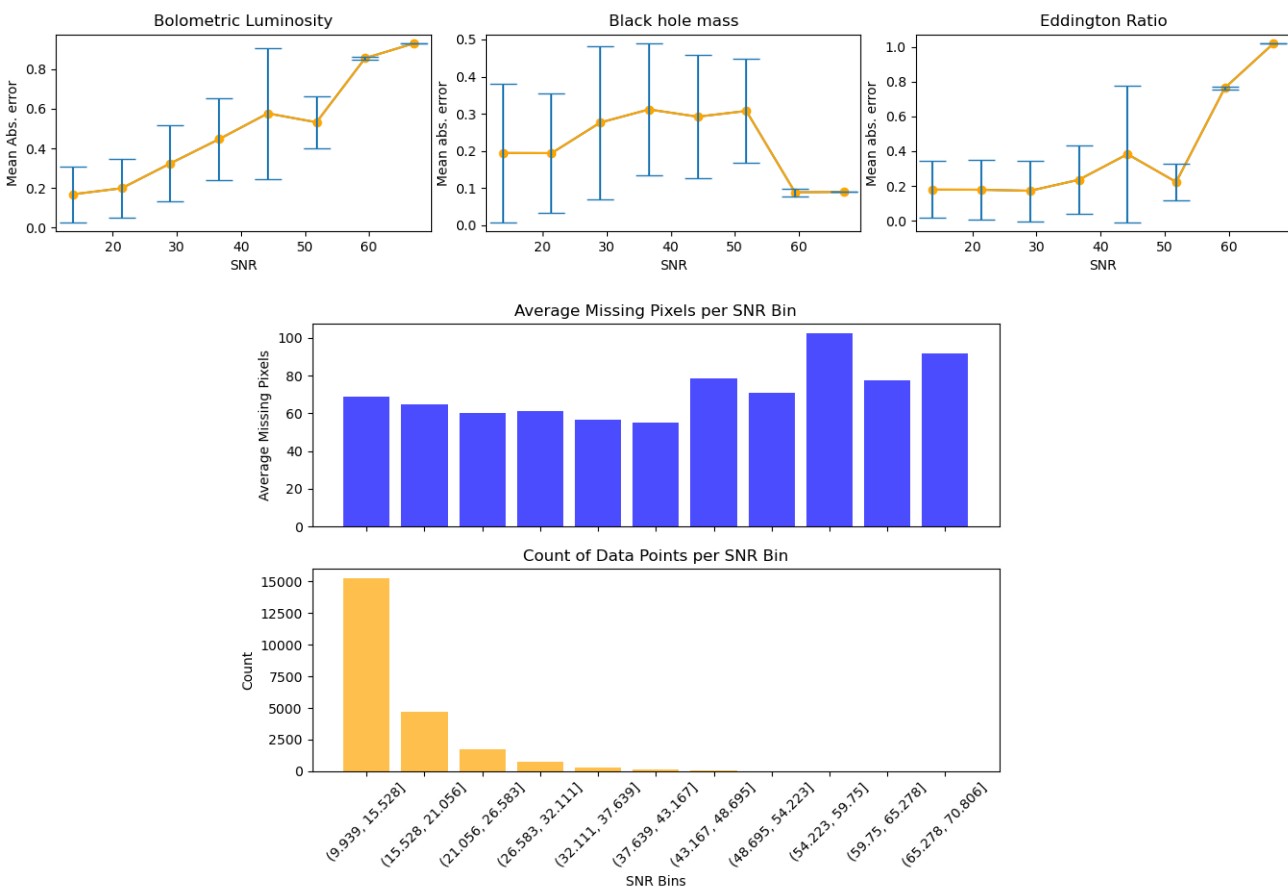

Figure 9: The SNR vs. error visualised with one standard deviation in each bin. The bottom plot shows the average number of missing pixels for the same SNR bins as well as the total number of quasars in each bin.

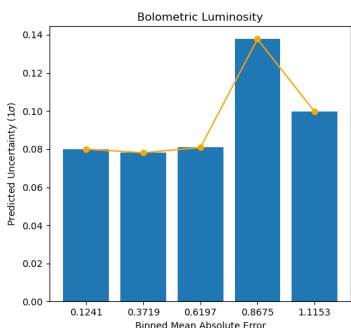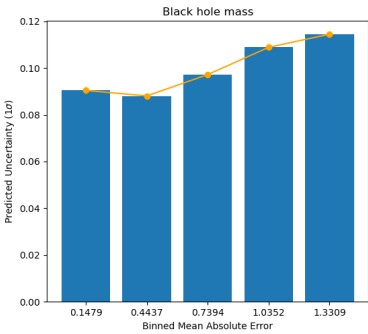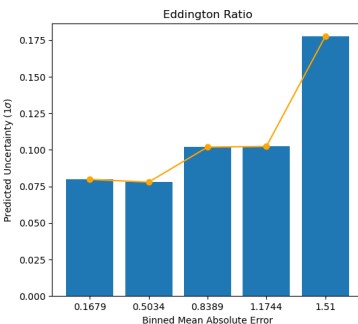

Figure 10: Uncertainty calibration across different error levels for each label computed over test data. The x-axis denotes centers of evenly spaced bins across the range of the MAE.

## F  Cross-validating $M$ and $Q$

The two main parameters of our shared framework which need to be fixed at the outset are: the number of inducing points $M$ and the latent space dimensionality $Q$. We set $M$ at 250 in all 22k experiments after cross-validation to $M = 1000$ and found that $M = 250$ gave the best possible trade-off in terms of speed and accuracy. The reconstruction results with $M = 1000$ were only marginally better than with inducing points $M = 250$, but the compute was significantly increased due to the cubic scaling in inducing points.

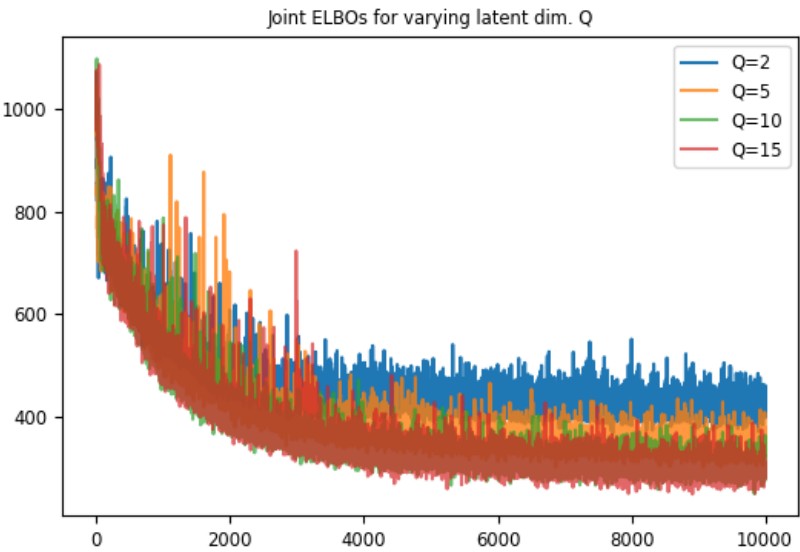

Figure 11: Sensitivity to Q: The negative ELBO objective for varying latent space dimensionality (lower is better)

In Fig. 11, we visualize the evolution of the ELBOs across varying latent dimensionality. We notice a meaningful improvement in increasing the dimensionality from $Q = 2$ but very marginal gains beyond $Q = 10$; we use this setting in experiments. It may be important to highlight that due to *automatic relevance determination* of the squared exponential kernel, setting a high latent dimensionality should not degrade results as the model automatically prunes redundant dimensions by driving the corresponding inverse lengthscales to 0. However, they do increase the compute cost, hence, it is important to set $Q$ at a reasonable value which is flexible enough for structure discovery and not too constrained, while simultaneously minimising the computational burden.

