# OpenReview forum: "Shared Stochastic Gaussian Process Latent Variable Models: A Multi-modal Generative model for Quasar spectra"
_TMLR — Accepted by TMLR_

### Review · Reviewer_qTSY · 2024-10-09

**Summary Of Contributions:**

This paper introduces a scalable generative model for quasars based on GPLVMs with SVI. The core technical observation is that "scientific labels" such as the black hole mass or Eddington ratio of a quasar and the quasar's spectra may be modeled with separate GP decoders using the same latent embedding as spectra, i.e., have a "shared" latent representation. The proposed method is shown to hold in general for observation spaces (X, Y), and then applied specifically to quasars, where it is shown that (1) the spectra are effectively captured in latent space, (2) the spectra may be reasonably interpolated, even with large amounts of missing data, and (3) the scientific labels may be estimated using only the spectra. These claims are evaluated on a large dataset consisting of ~22,000 astronomical measurements, and their scientific implications are discussed.

**Audience:**

Yes

**Claims And Evidence:**

No

**Requested Changes:**

## Critical
- (General) Regarding claims of scientific interpretation and significance, please temper claims or provide evidence, as discussed in the weaknesses section above.
- (Section 7.2) Please provide more direct evidence for the claim that the reconstruction accuracy of scientific labels is robust and independent of the SNR. In particular, please also comment on the relevance of the SNR > 10 cut to the data.
- (Figure 5) Please clarify what the uncertainty bands represent (i.e., $1\sigma/1.96\sigma/2\sigma$).

## Other Suggestions
- (General) All figures are a bit blurry. I think it would enhance the presentation to save figures in vector format or with a higher DPI.
- (General) Please consider incorporating the limitations discussed in Appendix B into the main text.
- (Section 1) The last sentence of the dimensionality reduction paragraph (i.e., "These methods however, are not generative [...]") was a bit difficult to parse -- I'd suggest rephrasing it for clarity.
- (Section 1) I'm not convinced of the claim that "generative latent variable models of late have supplanted traditional dimensionality reduction techniques"; e.g. the UMAP paper is recent and very highly cited, but is not a generative latent variable model. I would suggest using a word other than "supplanted" here, as this claim is not otherwise important to the paper.
- (Section 3) I think I understand that missing dimensions are algorithmically handled by simply not computing the corresponding gradients -- i.e. if the dimension $d^*$ is missing data point $x_*$, then the ELBO is updated as $\mathcal{L}_X \gets \mathcal{L}_X + \log p(z^{*}) + \sum_{d \neq d^{*}} \mathbb{E}_q \big[ p(x^{*}_{d} \,|\, f_d, z^{*}, \sigma_x^2) \big]$. This seems pretty clear by the use of `GaussianLikelihoodWithMissingObs` in the code, but explicitly highlighting this in Section 3 might improve clarity.
- (Algorithm 1) Semantically, the Training Framework and Prediction Framework parts of Algorithm 1 should probably be split into two separate algorithms, since they each have separate inputs and outputs.
- (Figure 6) Please update the caption of Figure 6 so that the content of the plot is self-contained, particularly since its description in Section 7.3 succeeds the plot itself.

## Minor Typos
I also noticed a few possible typos in the text, which I hope are useful to the authors.
- (Section 3) The second arrow in $x^* \to z^* \to p(y_l^* \, | \, z^*)$ is longer than the first.
- (Algorithm 1) In Step 2 of Prediction, "$\gets$" and "$=$" are used inconsistently.
- (Section 7.3) I suggest that "black hole masses in the range $\log_{10}(M_\cdot /  M_\odot) = 7.9-10.0$" should read "black hole masses $\log_{10}(M_\cdot /  M_\odot)$ in the range $[7.9, 10.0]$".
- (Section 2.1) Before Equation (4), there is a typo in "this yields".
# References
Eilers, A. C., Hogg, D. W., Schölkopf, B., Foreman-Mackey, D., Davies, F. B., & Schindler, J. T. (2022). A Generative Model for Quasar Spectra. _The Astrophysical Journal_, _938_(1), 17.

Lalchand, V., Ravuri, A., & Lawrence, N. D. (2022, May). Generalised GPLVM With Stochastic Variational Inference. In _International Conference on Artificial Intelligence and Statistics_ (pp. 7841-7864). PMLR.

**Strengths And Weaknesses:**

## Strengths
- The main strength of this work is in its application, which seems to be a novel and impactful place to apply recent GPLVM work. In particular, the improvement of using SVI over existing work in the astrophysics literature such as (Eilers et al., 2022) is apparent w.r.t. scalability.
- The technical contributions are clearly stated and developed.
- The provided code is well-written and self-documented, which I view as a big plus. To this end, the pseudo-code in Algorithm 1 is also very clear.
## Weaknesses
 - (General) Compared to (Lalchand et al., 2022), there isn't much technical novelty, mainly showing that sharing inducing points still leads to a factorized ELBO. This result is unsurprising, since the GPs between dimensions are still conditionally independent, just with different hyperparameters, and indeed, this is how the derivation proceeds.
   Nevertheless, this is a well-motivated approach to a novel application, and technical novelty isn't a criterion for acceptance, so this is not a big issue.
 - (General) Many of the claims made about the scientific significance of this work are not easily verifiable by someone in the machine learning community. In particular, some of the comments in Section 8 read as claims rather than speculation; they may be straightforward to an astronomer, but would not be known to the TMLR audience.

    For example, the final claim that "[the] new model allows us to now also use quasars as standard candles, [...] enabling new constraints on the dark energy content and Hubble constant [in the future]" seems like a strong claim, but has no further elaboration or reference. If this claim relies on the accurate reconstruction of the absolute luminosity, it seems very important to experimentally verify this.

   Similarly, it is claimed that "[the new method] allows us to determine the masses of SMBHs for quasars at all redshifts from single-epoch data and does not require coverage of specific emission lines, since it can handle missing data. This opens up [new possibilities...]." My non-expert understanding (based mostly on Section 7.1.1 of (Eilers et al., 2022) and the references therein) is that single-epoch estimators of the SMBH mass do exist, but are unreliable or inapplicable with high redshifts. But in the current manuscript, it doesn't seem like any results are evaluating the prediction quality of the SMBH mass w.r.t. redshifts, which seems central to the claim. Moreover, if the claim is that the proposed method reconstructs the black hole mass *more* reliably than existing methods, this should be verified.
- (Section 7.2) It is hard to judge the quality of uncertainty quantification here, especially with respect to black hole mass. It would be nice to see a calibration plot.
- (Section 7.2) Likewise, the claim that the error is effectively independent of SNR is not easy to verify from Figure 5. This claim is counterintuitive to me, and should be explicitly verified, e.g., by plotting the average error vs. SNR. When making this claim, it also seems important to note that a cut of SNR > 10 was applied to the data.
- (Appendix B) It's great that this work acknowledges some of its limitations in Appendix B, as well as some future directions to address them. I think it would strengthen the paper to include this brief discussion somewhere in the main text.
## Other Questions
- (General) The proposed method is not constrained to just (X, Y) -- is there a reason to consider all scientific labels to have the same kernel hyperparameters?
- (Code) It seems like `spectra_dims` in `baseline_gplvm` (which is only used in the `partition` option, and doesn't seem to appear in the paper) refers to parts of the latent space are given a "rougher" kernel, hoping this corresponds to the label outputs. Is this a correct interpretation? If so, the variable name `spectra_dims` may be a bit confusing/misleading, and it might help clarity to just get rid of this unused code.
- (Equation 12) Why is there a $\cdots$ in the integrals?
- (Section 2.1) Should the paragraph after Eq. (1) be referring to latent points $z, z'$ instead of $x, x'$?
- (Section 4) Why are there circular arrows for $f_d$ and $f_l$? And should both be in their own plates (over $d=1,\dots, D$ and $l = D+1, \dots, D+L$, respectively)?
- (Figure 5) The caption figure says the error bars denote one standard deviation, while the text says $2\sigma$. Looking at `visualization.py`, it seems like it may actually be $1.96\sigma$. Please clarify what is plotted, as this is crucial to interpreting the figure.

---

> ### Author Response · Authors · 2024-11-05
> **Response to Reviewer qTSY**
>
> I thank Reviewer qTSY for their detailed review and their time. I enclose my responses below:
>
> **Weaknesses**
>
> *Most of the points here have an accompanying bullet point in the other/critical suggestions. I have addressed a few points on the technical contribution in the general rebuttal above as well as rewording the astronomy claims.*
>
> I am happy to include a calibration plot for the black hole masses as well as the other labels to provid another view of the predictive quality.
>
> **Other Questions**
>
> $\cdot$ This is a good question, currently we maintain two copies of the covariance matrices as they are shared between the modalities. We can consider disentangling the kernel hyperparameters and learning 1+4 covariance matrices, this is tantamount to the version of the model having 5 modalities instead of 2. While it does seem intuitive that adding this flexibility of disentangling the decoders should sharpen the predictions but this is not what happens in practice. The GPLVM is conceptualised as a generative dimensionality reduction approach so it works by compressing a high-dimensional vector to a low-dimensional space, hence, it derives statistical strength from this sharing of hyperparameters and the latents (like in a VAE the amortisation with an encoder plays that part); in some sense we are using a single GPLVM jointly for the labels which is exactly what one would do if we didn't have any spectra and just had some high-dimensional features of scientific properties. If we were to work on a single label column, we would be turning the GPLVM on its head as we would then try to learn a mapping from a high-dimensional latent space to a single-output and while they can be used to do that and it isn't wrong (GP regression is a mapping from some high-dimensional output to scalar targets) it is not learning the latents with a view to learn a compressed representation. Currently, the latent space is still relatively high-dimensional (10) compared to the dimensionality of the labels (4), it is also jointly informed by very high-dimensional spectra. We have tried this version of disentanglement for the labels but it offers no advantage in terms of prediction and its actually somewhat worse, perhaps due to overfitting. But if you would like to include those results in the appendix we can add those in. Further, we normalise all outputs before training as is standard practice for a GPLVM.
>
> $\cdot$ Yes, that is correct, since the kernels factorise over dimensions and are closed over the multiplicative operation, one approach we tried was to see if we can gain anything by selecting different kernel functions for the two views of the data. As this doesn't change the inference framework ..just about adding more flexibility. In the end, we didn't see the real advantage in the combinations tried and more work would be needed to understand how to partition the dimensions. But clearly, this also provides a natural extension of the shared model we have described in the paper. I am happy to add a section to talk about extensions of the method..as this might also help garner more methodological interest.
>
> $\cdot$ The $\ldots$ in the integral is just to show that we are integrating out several function vectors (finite evaluations of GPs) each denoting a column $1....D+L$.
>
> $\cdot$ Section 2.1, agreed, this is a typo and the kernel inputs should be $\mathbf{z}$, and the kernel term should be $k\_{\theta}(\mathbf{z}, \mathbf{z}^{\prime})$.
>
> $\cdot$ The circular arrows just indicate that the values in the vector $\mathbf{f}\_{d}$ are correlated to each other since their prior has a full-rank covariance matrix given by $K\_{nn}$. This is a common protocol, also shown with self-referential arrows in the seminal paper Hensman et. al, 2013. Further, there should be a plate notation over the GPs which I think I erroneously skipped in the interest of clarity but decided to introduce it over the $N$ outputs, so it should also be shown for the GPs, I agree.
>
> $\cdot$ Upon checking the code, the plot (Fig. 5) denotes a 1\sigma interval and the text is incorrect, it should read 1\sigma interval and not 2\sigma. Apologies for the sloppiness here, but I remember trying to produce a plot where the vertical lines were not overwhelming the points and ended up downsampling and showing the 1\sigma bars instead of the typical 1.96\sigma I have done for the spectra.
>
> **Critical**
>
> $\cdot$ On scientific interpretation and significance: Since all the reviewers raised concern on this I have toned down the whole section and added a short paragraph on the need for further testing (included in the general rebuttal above). Have also included additional references to your point.

---

> > ### Author Response · Authors · 2024-11-05
> > **Response to Reviewer qTSY (Cont.)**
> >
> > $\cdot$ Agreed, it is straighforward to show that the error is uncorrelated with the SNR per data sample, with a error vs. SNR plot. I am happy to include that. The SNR > 10 is just a data quality cut, basically, it restricts the sample to slightly brighter targets where we can be assured of the robustness of the measurements. I can run an experiment with a higher data quality cut like SNR > 15, but the training data would be significantly smaller. The idea of SNR > 10 was really trade-off between yielding a high-quality and moderate sized dataset.
> >
> > $\cdot$ You are right, I was going for the 95% prediction interval as the marginals are Gaussian, hence, the $1.96\sigma$ for the error bars in the code; in the plots I seem to have denoted $2\sigma$ but I really mean $\approx 2\sigma$. Also in the text, I refer to the error bars as $2\sigma$ which I will change to 95% prediction interval.
> >
> > **Other Suggestions**
> >
> > $\cdot$ Sorry about that, I will try and change their resolution to make them sharper.
> >
> > $\cdot$ I have moved the Limitations to the main text.
> >
> > $\cdot$ I have broken this sentence down into two parts to make it easier to read.
> >
> > $\cdot$ Agreed, I think 'supplanted' maybe too strong a word as traditional dimensionality reduction like UMAP and t-SNE are still widely used and a whole range of unsupervised machine learning for clustering / structure detection like tasks is still underpinned by non-generative methods.
> >
> > $\cdot$ Agreed, I can describe the fact that the extended ELBO (to reflect the additional test point $\mathbf{x}^{\*}$ with missing dimension $d^{\*}$) can be written out as, $\mathcal{L}\_x^{\*} \gets \mathcal{L}\_x + \log p(\mathbf{z}^{\*}) + \sum\_{d \neq d^{\*}} \mathbb{E}\_q \big[ p(\mathbf{x}^{\*}\_{d} ,| \mathbf{f}\_d, \mathbf{z}^{\*}, \sigma\_x^2) \big]$.
> >
> > $\cdot$ I have split Algorithm 1 into a training algorithm and a prediction algorithm.
> >
> > $\cdot$ I have updated the caption of Figure 6 which summarises the experiment on cross-modal prediction, where we predict spectra corresponding to simulated scientific labels. We do this one at a time, hence, to understand how the spectra is influenced by black hole mass we keep all the other labels fixed to their training mean values.
> >
> > **Minor Typos**
> >
> > Thank you, these have all been addressed.
> >
> > J. Hensman, N. Fusi, and N. D. Lawrence. Gaussian processes for big data. Proceedings of the 29th Conference on Uncertainty in Artificial Intelligence (UAI), 2013

---

> ### Comment · Reviewer_qTSY · 2024-11-07
> **Response to the Authors**
>
> Thank you for your thoughtful replies, which have addressed several of my concerns. In particular, I am satisfied with the changes made to the Scientific Impact section. A few concerns remain, which I list below:
>
> - (On Novelty) I would maintain that the result is rather unsurprising mathematically -- even if nontrivial to implement -- but also recognize this is not a criterion for acceptance at TMLR. I think the suggestion is more of contextualization; the fact that a (mathematically) simple modification is helpful in an application can even be viewed positively, and as Reviewer T1nP pointed out, can aid in simplifying the exposition and clarifying contributions.
> - (On Separate Hyperparameters for Each Label) Thank you, this is an insightful answer. I recognize now that there is a tradeoff between providing expressivity in the generative model and learning an informative enough embedding. Indeed, I think it would strengthen the manuscript significantly to include this experiment and your provided discussion surrounding it.
> - (On Mixed Kernels) Sure, I think discussion surrounding methodological future work is a worthwhile addition. If you decide to keep the code, I'd suggest at least documenting this approach there.
> - (On $\cdots$ in the Eq. (12)) I would argue that integration over each variable is already implied by the notation -- e.g., this is why the dots don't appear in Eq. (13). Though ultimately this is a small point, the meaning is still obvious either way.
> - (On the Bayesian Plate Diagram) Thanks for the clarification. I don't see this updated in the revision, however. Also, I think a note about the self-referential arrows in the caption would be useful -- even if this notation is standard in the GPLVM literature, I don't find it standard in the greater Bayesian literature.
> - (On Including Limitations) Thank you for including limitations in the main text, as previously mentioned, I think this improves the presentation. That said, Appendix B is still present -- please remove it, now that its contents appear in Section 8.
> - (On SNR) Yes, please do include this plot. Also, I understand that the SNR cut is a data quality cut, but this is what makes the claim so counter-intuitive in the first place: reconstruction quality being independent of SNR implies high-quality reconstructions on low-quality data. I would consider it a reasonable but substantively different claim to say "the reconstruction quality is independent of SNR after data achieves a baseline level of quality, as quantified by an SNR of at least 10."
> - (On the word "supplanted") Thanks. "Supplanted" still seems to appear in the revised version, please remember to change it.
>
> Some typos:
> - "A few hundred Myr (million) old" should be "a few hundred Myr (million **years**) old" (emphasis mine). Also, since Myr isn't otherwise used, consider just using "million years" and avoiding the jargon.
> - "Additionally, we have shown that our model is able to predict **also** other physical properties" has an extraneous "also."
> - In "[...] new opportunity to use quasars as so-called **“standard candles” ."** the period should appear inside of the quotation marks, and there is an extra space.

---

> > ### Author Response · Authors · 2024-11-18
> > **Response to Reviewer qTSY (post rebuttal)**
> >
> > Thank you for reviewing the changes. The current revision uploaded includes the following changes:
> >
> > $\cdot$ I have included a new section which talks about methodological extensions as recommended.
> >
> > $\cdot$  Bayesian plate notation: I have added the plates to denote that there are D GPs $f_{d}$ and $L$ GPs $f_{l}$ . For clarity, I also change the index of the $\ell$ GPs as going from $D+1, \ldots, D+L$ to avoid ambiguity, as I need the second observation space to have $L$ dimensions rather than $L-D$ dimensions. Upon further reflection, I have concluded it is not absolutely necessary to have the self-referential arrows as while it is present in some papers, I agree it is not standard practice, I have removed these and updated the figure.
> >
> > $\cdot$ Eq. 12: I agree, it is inconsistent to have the $\ldots$ in  Eq. 12 and not in Eq. 13. I have corrected this.
> >
> > $\cdot$ Removed duplicate section 'Limitations' from the Appendix.
> >
> > $\cdot$ SNR Experiment: I agree with this, I have caveated the claim in the paper and re-iterate that we apply a data quality cut of SNR> 10.
> >
> > $\cdot$ I have reworded the sentence with the word 'supplanted'.
> >
> > Typos:
> >
> > Thank you - fixed all these as requested.
> >
> > Pending are the additional experiments:
> >
> > 1. SNR v. error plot.
> > 2. Calibration plot for the labels.
> >
> > I will be including these in an updated revision by eod 19th.
> >
> > I will certainly need a bit more time to sharpen all the figures and correct their legends as I would have to reproduce all of them. As well as the section on the disentanglement experiment (separate hypers for each output label) which is straightforward to implement.

---

> > > ### Author Response · Authors · 2024-11-19
> > > **Validation experiments enclosed in latest revision**
> > >
> > > Dear Reviewer qTSY,
> > >
> > > I have included the two additional validation experiments proposed in Appendix C and D.
> > >
> > > 1. SNR v. error plot.
> > > 2. Uncertainty calibration experiment for the labels.
> > >
> > > Thank you.

---

> > > > ### Comment · Reviewer_qTSY · 2024-11-21
> > > > **Thanks for Additional Plots**
> > > >
> > > > Thank you for including these!
> > > >
> > > > Regarding the SNR vs. error plots, the plots are a bit difficult to read because of the density of points. I think it would be very helpful to include mean + variance (perhaps interpolated by a GP, or sliding window, etc.). This would also help clarify that the triangular shape is an artifact of the data distribution. As discussed in the appendix, there also (visually from the scatterplot, at least) seems to be a weak relationship. As long as this is mentioned in the main text (and the appendix referenced), I think I'm satisfied with these additional experiments.
> > > >
> > > > Regarding calibration curves, I am much more used to seeing the style of calibration curve implemented in, e.g., the uncertainty toolbox package in Python [1], and would recommend these be included in a final version, though I do not view this as critical.
> > > >
> > > > ## References
> > > > [1] https://uncertainty-toolbox.github.io/tutorial/

---

> > > > > ### Author Response · Authors · 2024-11-29
> > > > > **Response**
> > > > >
> > > > > I have added a bit more analysis on the SNR vs. error. Your suggestion was spot on.
> > > > >
> > > > > When you visualise the binned error for different SNR levels, there seems to be an unusual positive correlation (albeit weak) for luminosity and the Eddington ratio. I looked into it more carefully and it seems like there are two effects at play here.
> > > > >
> > > > > The unusual weak positive correlation can be better understood in the context of the count and missing pixels analysis in each SNR bin. First, there are very few high SNR (> 40) quasars in the dataset and further, these quasars also have more missing pixels than quasars at lower SNRs. Since the training dataset is biased towards lower SNR quasars, the model generalises poorly to higher SNR examples, for luminosity and Eddington ratio. It is highly likely that the pattern of missing pixels is similar among brighter objects and in turn among less luminous objects, making it difficult for the model to adapt to this shift at test time due to the acute data imbalance.
> > > > >
> > > > >  | SNR_Bin  | Count | MissingPixels
> > > > >
> > > > > 0  |  (9.939, 15.528]  | 15247 |  68.95
> > > > >
> > > > > 1  |  (15.528, 21.056]  |  4696 | 64.66
> > > > >
> > > > > 2   | (21.056, 26.583]  | 1724 | 59.98
> > > > >
> > > > > 3  | (26.583, 32.111]   | 717 | 61.17
> > > > >
> > > > > 4  | (32.111, 37.639]   | 265 |  56.43
> > > > >
> > > > > 5   | (37.639, 43.167]  |  120 | 55.02
> > > > >
> > > > > 6  | (43.167, 48.695]    | 49 | 78.46
> > > > >
> > > > > 7  | (48.695, 54.223]   |  16 |  70.81
> > > > >
> > > > > 8   | (54.223, 59.75]   |   6 |  102.33
> > > > >
> > > > > 9   | (59.75, 65.278]   |   3 |  77.34
> > > > >
> > > > > 10  | (65.278, 70.806]   |  1 | 92.00
> > > > >
> > > > > I have included this data as a plot in the revision on the appendix section "Sensitivity to SNR".

---

### Review · Reviewer_T1nP · 2024-10-17

**Summary Of Contributions:**

The paper proposes an extension of the stochastic variational Gaussian process latent variable model (GPLVM) (Lalchand et al. 2022) to the multi-view setting.

The application is in astrophysics - specifically, the authors consider very luminous galaxies, for which they have both spectral pixel data and tabular data measurements such as black hole mass, accretion rate, and luminosity.

### Model

In GPLVM, the coordinates of the data $x = (x_1,\dots, x_D)$ are modeled independently. That is,
$$p(x|f, z) = \prod_{d=1}^D N(x_d; f_d(z), \sigma^2)$$
where $f_d$ are GPs and $z$ is the latent representation of $x$. Because the coordinates are modeled independently, the GPVLM model naturally can incorporate another data view $y=(y_1,\dots, y_L)$, simply by concatenating the data $(x_1,\dots, x_D, y_1,\dots, y_L)$ and adding additional GPs $f_l$, $l=1,\dots, L$. This is what this paper proposes (where the $f_l$ are allowed different kernel hyperparameters than $f_d$).

For optimization, the paper follows Lalchand et al. 2022. Specifically, the paper uses the "point" version of the Lalchand et al. 2022 algorithm, which obtains point estimates for the latents $z$.

The authors conduct extensive experiments demonstrating:
- on test data, the model reconstructs quasar properties similarly or better than the Lalchand et al. 2022 model
- on test data, the model reconstructs quasar spectra well even when the data is partially unobserved
- on test data, the model predicts quasar properties well from only the quasar spectra
- the model can generate example spectra for given scientific properties

### References
- Vidhi Lalchand, Aditya Ravuri, and Neil D. Lawrence. (2022) Generalised GPLVM with Stochastic Variational Inference. Artificial Intelligence and Statistics.

**Audience:**

No

**Claims And Evidence:**

Yes

**Requested Changes:**

### Critical changes
- To my understanding, the model is a very simple extension of Lalchand et al. 2022 (see "summary of contributions"). In particular, the model is the same as Lalchand et al. 2022, just with different hyperparameters for the different data dimensions. As such, the paper should be much clearer about this.  This would also improve exposition of the model.
- eq. 15 and eq. 16: this is confusing - in one, you integrate over $f$, in the other, you don't. (see also comments in weaknesses)
- Can the authors consider the MAP and Bayes version of stochastic variational GPLVM?
- Fig 1: I'm confused by the self referential arrow on the $f_d$ - aren't the coordinates conditionally independent given the data?
- Experiments: some paragraphs are a little confusing for readers without astronomy knowledge. For example: "Note that we omit the reconstruction of the redshift label z, as we do not expect any redshift evolution in the spectral shape of quasars, but rather the GPLVM is likely learning where spectral pixels are measured vs. missing. Since we are using data from SDSS, all spectra have the same observed wavelength coverage, which is then shifted to the rest-frame wavelengths by dividing by (1 + z), and thus the observed pixel dimensions in our data set contains the redshift information, rather than any spectral differences." As a non-astronomer, I do not understand this. Why do you not expect redshift evolution in spectra? What are the rest-frame wavelengths and why are they shifted? You then say that pixels contain redshift information, but earlier you said that you don't expect redshift evolution in the spectra pixels?
- Table 2: can you compare to Lalchand et al. 2022 too here?
- p. 14: "Our new model allows us to determine the masses of SMBHs for quasars at all redshifts from single-epoch data and does not require coverage of specific emission lines, since it can handle missing data." How does the model handle all redshifts? What if there are redshifts missing in the training dataset?
- p. 2: "We don't employ this setting as it is not straightforwardly applicable to missing data contexts." Masked autoencoders are often used to learn representations - can they not be applicable here? Moreover, in multimodal settings, autoencoders are now regularly used to predict one modality from another modality. Can the authors speak more to why they only consider GPLVMs, and not another VAE-based multi-view model?


### Other changes
- I'm confused by the inducing point distribution $u_d|\widetilde{Z} \sim N(u_d; 0, K_{mm})$. Is the kernel computed with respect to $\widetilde{Z}$? This should be made clear.
- eq. 13: what happened to conditioning on $Z$?
- Experiments: can you say "test" $X^*, Y^*$ instead of "unseen"?

### Typos
- p. 2: ">400 x bigger" re-write: datasets over 400 times larger
- p. 3: "Essentially by keeping the representation of $u_d$ uncollapsed." This sentence isn't complete and is also confusing - in what sense is $u_d$ uncollapsed and why does that help?
- p. 4: equation (3) seems superfluous as it was derived in Lalchand et al. 2022
- p. 4: "yiels" -> "yields"
- p. 4: "we introduce the idea of an additional observation space..." this could be made more clear e.g. "we extend this model to accommodate an additional data view"
- p. 6 "variaitonal" -> "variational"
- change your `\citep` to `\citet` when for in-text references
- p. 14: "few hundred Myr old" - million years?

**Strengths And Weaknesses:**

## Strengths

The paper considers an important application area (multi-view astronomical data). The model seems well suited for this data and can lead to interesting scientific results in astronomy, as demonstrated by the extensive experiments in the paper.

## Weaknesses

- To my understanding, the model is a very simple extension of Lalchand et al. 2022 (see "summary of contributions"). In particular, the model is the same as Lalchand et al. 2022, just with different hyperparameters for the different data dimensions. As such, the paper should be much clearer about this. As a result, the paper may be better suited to an astronomy focused journal.
- Some of the exposition of the model is confusing. For example, in eq. 14, the GPs $f_d$ are integrated out. But then in eq. 16, the actual objective, the expectation is taken with respect to the variational distributions. Further, eq. 16 is not evaluated wrt to this expectation. To understand the ultimate objective function, I had to go back to Lalchand et al. 2022 (e.g. their eq. 6).
- Lalchand et al. 2022 consider point, MAP and Bayesian options for estimating $z$. In many cases, they find the Bayesian version (i.e. specifying a variational distribution $q(z)$ and learning the variational parameters) has better reconstruction. Why did the authors not consider this version?
- p. 2: "We don't employ this setting as it is not straightforwardly applicable to missing data contexts." Masked autoencoders are often used to learn representations - can they not be applicable here? Moreover, in multimodal settings, autoencoders are now regularly used to predict one modality from another modality. Can the authors speak more to why they only consider GPLVMs, and not another VAE-based multi-view model?
- Is the per-column conditional independence assumption on $x$ appropriate for pixel data?
- Some of the details in the experiments are difficult to understand as a non-astronomer (see requested changes).

---

> ### Author Response · Authors · 2024-11-04
> **Response to Reviewer T1nP**
>
> I would like to thank Reviewer T1nP for their detailed review and their time. I enclose my responses below:
>
> I address the points on technical novelty in relation to Lalchand et al, 2022 and the scientific claims section in the general response above.
>
> **Weaknesses**
>
> The weaknesses mentioned are all connected to critical changes requested, hence, I provide the pointer to the answers in the critical changes section.
>
> $\cdot$ On novelty - Addressed in the general rebuttal.
>
> $\cdot$ On confusing exposition - related to pointer 2 in critical changes.
>
> $\cdot$ On other modes of inference like MAP and full Bayes - related to pointer 3 in critical changes.
>
> $\cdot$ On the use of VAE-based multi-view models - related to pointer 8 in critical changes.
>
> $\cdot$ On the per column independence assumption for spectral dimensions.
>
> This is a very relevant point. It is perfectly valid to ask, why not model the output dependencies between all dimensions ($D+L$) of all the observation spaces $X$ and $Y$.
>
>  **Logic for current architecture** The shared latent space ($Z$) is the only bridge between the two observation spaces as the decoders are independent. In our view, the shared space $Z$ serves as a regulariser to the independence of the decoders. However, the models is not just two independent GPLVMs stacked together as then it would not be possible to generate the spectra ($X$) from only observing the labels ($Y$) and vice-versa. This is an important goal. Ultimately, our design allows that because the inducing inputs and latents are shared between all the GP decoders across both the observation spaces.
>
> There are models inliterature where the  inter-output dependencies are additionally modelled by a free-form matrix. These come under the 'linear' model of coregionalisation or the 'intrinsic' model of coregionalisation. Upon further reflection, some of the features we lose when we model output dependencies are:
>
> Factorisation: The ELBO objective would no longer factorise across dimensions ($D$ or $L$) within each observation space (it does in our model). However, since it would still factorise across $N$ -- this may not be so catastrophic if $D$ is not very large (less than 200). But in astronomy, $D$ can be very large as the spectra pixels may be as large as 2000, this makes the coregionalisation model very compute internsive. It is not very clear if we can learn that many free-form model parameters with SVGP.
>
> Compute and free-form hypers: Right now, we have very few model hyperparameters (3 + 3 across all $f_{d}$s and $f_{l}$s) which are updated in each iteration of the mini-batch. The correlated model would require all the free-form hypers to be updated in each mini-batch step (as they are global) slowing down the training significantly. We will also need more inducing points as SVGP is applied to vec$(f_{1:D}) \sim \mathcal{N}(0,K_{\text{coreg}})$ where $K_{\text{coreg}}$ is of size $ND \times ND$.
>
> Ultimately, we want a generative model which can simultaneously generate both spectra and labels to a high degree of fidelity, hence, a shared latent space is a very reasonable approach. If we can get further accuracy by modelling inter-output covariances i.e. explicitly correlating all the $f_{d}$s and all the $f_{l}$s - the additional costs may be justfied, but we really need to investigate (through experiments) if a shared latent space with correlated decoders is flexible enough to generate both the views of the data $X$ and $Y$ simultaneously.
>
> In our model we have covariance sharing (but skipped on dependency modelling) between all spectral pixel dimensions and between all the labels (the mappings are drawn independently), the implicit assumption is that the covariance structure does not change too much across the dimensions. Finally, one can view the whole model design as a trade-off between flexibility and regularisation.
>
> $\cdot$ On clarity of some statements in the experiments - related to pointer 5 in critical changes.
>
> **Critical Changes**
>
> $\cdot$ Addressed in the general rebuttal.
>
> $\cdot$ *Clarity in the exposition of the lower bound*. Eq. 15 is an equality, it is just the canonical GP marginal likelihood with two data views which conviniently factorises across the views, whereas Eq. 16 is a lower bound and can be directly derived as a sum of lower bounds for the two summation terms in Eq. 15.  It is a direct application of Eq. 6 in the single data view case.
>
> As for the confusion about the integration over $\mathbf{f}\_{d}$, we also integrate out $\mathbf{f}\_{d}$ in the lower bound as the expectation (in Eq. 16) is wrt the variational distribution which over all unknows (Eq. 2)$\prod\_{d=1}^{D}p(\mathbf{f}\_{d}|\mathbf{u}\_{d}, Z)q(\mathbf{u}\_{d})\prod\_{n=1}^{N}q(\mathbf{z}\_{n})$. However, I haven't explicitly shown the integration but write it as an expectation around the likelihood ..which has $\mathbf{f}\_{d}$ after the conditional. Will improve the clarity on this.
>
> (Cont. below)

---

> > ### Comment · Reviewer_T1nP · 2024-11-20
> > **response**
> >
> > Thanks for your detailed reply - I am satisfied with your answer to my per-column independence comment (indeed, this per column independence is also what VAEs do implicitly).
> >
> > I'm mostly happy with the clarifications you've made to the GPLVM derivations - thank you for implementing them.  However, I still have a few requests:
> > - personally, I think eq 3 should be omitted - for a TMLR paper on GPs, you can write the ELBO without derivation and just have a reference to Lalchand et al. 2022. The current eq. 3 and derivation is too detailed and looks a little messy
> >
> > - instead of the notation $\langle \cdot \rangle_{q(\cdot)}$ can you make it clear which distributions you are taking expectations with respect to? e.g. $\mathbb{E}_{p(f|u,z)q(u)}$. This will make things clearer, especially as you are not using a variational distribution for $f$.
> >
> > - Thank you for your helpful explanations about redshift and wavelengths. However, this paragraph is still very confusing to me (as a non astrophysicist/cosmologist) - I think it would be helpful to re-write this, incorporating some of your explanations to me.
> >
> > "Note that we omit the reconstruction
> > of the redshift label, as we do not expect any or only very little redshift evolution in the spectral shape
> > of quasars (as shown previously for large samples of quasar spectra by Yang et al. (2021); Onorato et al.
> > (2024)), but rather the GPLVM is likely learning where spectral pixels are measured vs. missing. Since we
> > are using data from SDSS, all spectra have the same observed wavelength coverage, which is then shifted to
> > the rest-frame wavelengths by dividing by (1 + z), and thus the observed pixel dimensions in our data set
> > contains the redshift information, rather than any spectral differences."
> >
> > Here is my attempt to understand so you can see the level of the average ML reader with no astro background:
> >
> > "Note that we do not attempt to reconstruct $Y_{redshift}$ from the spectra $X_{spectra}$. This is because the spectra have already been normalized to account for their redshift (that is, the spectra were shifted to rest-frame wavelength space). Consequently, the normalized spectra shapes should not contain any information about their redshift (Yang et al. (2021); Onorato et al. (2024)). Although the normalized spectra shapes do not contain redshift information, we still include $Y_{redshift}$ in our multimodal GPLVM. This is because in our experiments, excluding $Y_{redshift}$ marginally weakens the results, both in terms of spectra reconstruction and label prediction. We hypothesize this is because although the spectra shape do not contain redshift information, the missingness pattern of the spectra pixels is still informative of redshift.

---

> > > ### Author Response · Authors · 2024-11-28
> > > **Response to Reviewer T1nP (round 2)**
> > >
> > > Thank you for your suggestions and your approval of the other corrections, I enclose the following amendments.
> > >
> > > $\cdot$ I have removed Eq. 3 and referred Lalchand et al (2022) but left the latter forms just for clarity.
> > >
> > > $\cdot$ I have expanded the variational distribution in the expectation for Eq. 5, Eq. 15 and every else they appear in the paper.
> > >
> > > $\cdot$ Thank you for providing a guiding passage, it is indeed much clearer. I have re-worded it slightly and replaced it in the revision.
> > >
> > > **Not yet uploaded**
> > >
> > > I will be uploading a further revision experiment on including NLPD values for the baseline experiment for the two modes of prediction with spectra only and full test observation in the forthcoming one later in the day. Thank you.

---

> > > > ### Author Response · Authors · 2024-11-29
> > > > **Response to Reviewer T1nP (round 3)**
> > > >
> > > > I have now updated Table 2 with NLPD values (mean across splits) for both the baseline and shared models under both protocols (fully observed and spectra only). Earlier I had reported median values without the SE, the main reason for that was that I noticed some runs had outliers and chose the median as it is more resistant to it. Now I have re-run everything and found no extreme values as I noticed that the issue with outliers earlier was to do with the convergence of each run which had to be carefully checked. I will add a note about convergence checking in the appendix.

---

> > > > > ### Comment · Reviewer_T1nP · 2024-12-02
> > > > > **response**
> > > > >
> > > > > Thanks for all your changes. I only have some minor edits regarding writing:
> > > > >
> > > > > p. 3
> > > > >
> > > > > - this is not a proper sentence: "Essentially, by keeping the representation of u uncollapsed." use a semi-colon to connect with previous sentence, or re-write both sentences
> > > > >
> > > > > - lots of citeps instead of citets - please check this throughout the paper
> > > > >
> > > > >
> > > > > p. 4
> > > > >
> > > > > - can the authors be consistent with their expectation notation? e.g. in eq. 3 it is $\mathbb{E}$. in eq 5 it is $\langle \rangle$.
> > > > >
> > > > > "In the next
> > > > > section, we introduce the idea of an additional observation space with L dimensions and how they can be
> > > > > modelled by their own stack of independent GPs fl and learn their own set of hyperparameters for additional
> > > > > flexibility but share the latent embedding Z and inducing inputs
> > > > > Z to model correlations between the different
> > > > > output spaces."
> > > > >
> > > > > This sentence is too long - break it up to be clearer for readers.
> > > > >
> > > > > p. 6
> > > > >
> > > > > formatting issue - need a space before $\forall l...$
> > > > >
> > > > >
> > > > > p. 9
> > > > >
> > > > > can you change the name of disentangled? maybe "specific" to stand for label-specific hyperparameters? "disentangled" is an overloaded term in representation learning and may prime readers to think of something different.
> > > > >
> > > > > p. 13
> > > > >
> > > > > "We hypothesize that
> > > > > while the spectral shapes themselves do not carry redshift information, the patterns of missing may still
> > > > > encode some redshift-related information."
> > > > >
> > > > > missing word after "missing". pixels? (also this reads much better, thank you)
> > > > >
> > > > >
> > > > > - i don't think NLPD is defined anywhere? (based on a CRL + F search)
> > > > >
> > > > >
> > > > > "From Table 2 we can glean that the variability of the NLPD estimates is much higher under the cross-modal
> > > > > protocol of decoding scientific labels on the basis on the spectrum. As a result of which the marginal
> > > > > advantage of the shared model over the baseline model in terms of uncertainty quantification is more subdued."
> > > > >
> > > > > - these sentences need a re-write for clarity. much higher than what? e.g. "on test data, reconstructing scientific labels from spectra alone has a much higher uncertainty than reconstructing scientific labels from both modalities."
> > > > >
> > > > > - "as a result of which" - "of which" can be omitted
> > > > > - "more subdued" - clearer langugage is better. "when reconstructing scientific labels based on spectra alone, the shared model performs similarly to the baseline model. "
> > > > >
> > > > > - "It is also important to note that under the much harder
> > > > > cross-modal protocol, the performance of the shared model is only marginally weaker than the baseline model
> > > > > with the fully observed protocol."
> > > > >
> > > > > i don't think this is true, it seems in between baseline (fully observed) and baseline (spectra observed) for the most part
> > > > >
> > > > > p. 15
> > > > >
> > > > > - "(represented
> > > > > as NaNs in the data)"
> > > > >
> > > > > omit this (this is standard)

---

> > > > > > ### Author Response · Authors · 2024-12-04
> > > > > > **Response**
> > > > > >
> > > > > > Thank you for pointing these out, I will be sure to address them in the final version.

---

> ### Author Response · Authors · 2024-11-05
> **Response to Reviewer T1nP (Cont.)**
>
> **Critical Changes**
>
> $\cdot$ The shared latent framework is compatible with learning a full variational distribution $q(Z)$ over the latent variables. Capturing uncertainty in the latent space is crucial when we expect the latent variables to be less well-determined - this can certainly be the case when there are large amounts of missing data. Further, variationally integrating out the latents by learning a full posterior $q(Z) \approx p(Z|X,Y)$ is more robust to overfitting and yields qualitatively better uncertainty. In our work, we were mainly concerned by the ability to jointly generate valid spectra and labels post training - we did have missing dimensions but they were a small fraction of the complete observed dataset. In experiments we did not see a significant improvement in reconstruction quality, however, the uncertainty captured by the test log-likelihood was marginally better. Computationally, learning $q(Z)$ adds variational parameters per data point, $\lbrace \mu_{n}, S_{n}\rbrace_{n=1}^{N}$, $S_{n}\in \mathbb{R}^{Q}$.
>
> $\cdot$ Table 2 prediction of properties in the baseline model. I can include this benchmark in the revised version.
>
> $\cdot$  I apologise for the confusion here.
>
> **On redshift evolution** In astronomy, the term redshift refers to how light stretches to longer (redder) wavelengths as an object moves away from us, a phenomenon due to the expansion of the universe. For quasars, astronomers generally observe a consistent shape in their spectral features, regardless of distance. This consistency means that the fundamental shape of the quasar’s spectrum doesn’t change with redshift; only the position of features along the wavelength axis shifts as we look further away. This is why we don't expect "redshift evolution" in the spectral shape of quasars.
>
> **Observed vs. Rest-Frame Wavelengths** Observed wavelengths are what we detect with our telescopes on Earth, which get shifted by the redshift. The rest-frame wavelength refers to the wavelength as it would be observed if the object (quasar) were stationary relative to us. To get from observed to rest-frame, the wavelengths are "shifted" back by dividing by $(1+z)$, where
> $z$ is the redshift.
>
> The location of the observed pixels (which pixels have a value and which are missing) contain redshift information, but the value of the pixels (the measurements themselves) do not. I can clarify this further in the manuscript.
>
> $\cdot$ The redshift information is modelled like any other scientific label in our model (it is one of the 4 columns being modelled for the label view of the data). We have run experiments where we omit this column altogether and this marginally weakens the results, both in terms of spectra reconstruction and label prediction (as the position of pixels in these spectra -- and not their values -- inherently includes information about the redshift).
>
> $\cdot$ Yes, certaintly, masked autoencoders adapted for multiple views would be a suitable benchmark. This work really arose from finding a methodology to improve on the scalability of the analysis done in Eilers, 2022. The main advantage of using GPLVMs over deep models (also stated in Eilers, 2022) is robust quantification of uncertainty in the outputs, we expect to see higher epistemic uncertainty in dimensions where there is a higher fraction of missing pixels (Fig. 4 and 13). GPLVMs also lend themselves to modelling with heterogeneous noise (although we assume homoskedastic noise in this work), this would be a natural extension of the model. In some sense selecting a prior, kernel and noise framework as we do in the GPLVM makes the whole model more interpretable. In GPs the output uncertainty is obtained by design as opposed to autoencoder like models where the output uncertainty is not a first class concern and as such I don't believe would capture epistemic uncertainty in the way that GPs respond to missing data. However, this is still just an educated guess, and I am happy to include a MAE benchmark for atleast one experiment in the manuscript.

---

> > ### Author Response · Authors · 2024-11-05
> > **Response to Reviewer T1nP (Cont. Part 2)**
> >
> > **Other Changes**
> >
> > $\cdot$ Yes, the inducing variable distribution is computed on the inducing inputs denoted as $\tilde{Z}$. The covariance matrix $K\_{mm}$ is evaluated on the inducing inputs $[\tilde{z}\_{1}, \ldots, \tilde{z}\_{M}]^{T}$ which are global variational parameters and are learnt during mini-batch training.
> >
> > $\cdot$ Yes I agree, the notation seems inconsistent there, the inputs $Z$ should either be conditioned on in every mention of the likelihood or ommitted as implicit, but since I choose to write the likelihood with explicit conditioning on the inputs, it should be there in  Eq. 13 and 14 as it reappears in Eq. 15. I have changed as below:
> >
> >  $= \int\_{1:D} p(X|\mathbf{f}\_{1:D}, Z)p(\mathbf{f}\_{d}|\theta\_{x}) d\mathbf{f}\_{1:D} \int\_{D+1:L}  p(Y|\mathbf{f}\_{D+1:L}, Z)  p(\mathbf{f}\_{l}|\theta_{y})d\mathbf{f}\_{D+1:L}$ (Eq. 13)
> >
> >  $= \prod\_{d=1}^{D} p(\mathbf{x}\_{d}|\theta\_{x},Z) \prod\_{l=D+1}^{L} p(\mathbf{y}\_{l}|\theta\_{y},Z) = \prod\_{d=1}^{D} \mathcal{N}(\mathbf{0}, K\_{nn}^{(d)} + \sigma^{2}_{x}) \prod\_{l=D+1}^{L} \mathcal{N}(\mathbf{0}, K\_{nn}^{(l)} + \sigma^{2}\_{y})$ (Eq. 14)
> >
> > $\cdot$ Agreed, 'unseen' changed to 'test' for $X^{\*}, Y^{\*}$.
> >
> > **Typos**
> >
> > Thank you for pointing these out, they have all been fixed in the revised version. I would like to address pointers 2 and 3 in your list above:
> >
> > $\cdot$ p.3 on the uncollapsed representation of $\mathbf{u}\_{d}$: Canonical sparse GPs and the inducing variable formulation comes from Titsias, 2009 where the optimal inducing variable distribution $q^{\*}(\mathbf{u})$ is computed directly and plugged back into the ELBO (rendering the ELBO unfactorisable over N and hence, not compatible with very large data) which is now collapsed wrt $\mathbf{u}$ as it is just a function of the hypers. and inducing inputs. Whereas stochastical variational GPs, introduced in Hensman et al, 2013 were reformulated for very large data and this necessitates learning $q^{\*}(\mathbf{u})$ numerically, through stochastic optimisation of the lower bound (hence $q(\mathbf{u})$ is represented in the bound and is uncollapsed), the mean and covariance are variational parameters which are \emph{learnt} through mini-batched training.
> >
> > $\cdot$ p.4 Equation 3 was just included to ensure a stand-alone background section and I supposed that showing the construction of the rudimentary ELBO in Equation 4 would help provide a foundation for understanding Equation 6 and eventually Equation 16. But I am happy to move it to the appendix or explicitly call it 'Background'
> >
> > **References**
> >
> > J. Hensman, N. Fusi, and N. D. Lawrence. Gaussian processes for big data. Proceedings of the 29th Conference on Uncertainty in Artificial Intelligence (UAI), 2013
> >
> > Vidhi Lalchand, Aditya Ravuri, and Neil D. Lawrence. (2022) Generalised GPLVM with Stochastic Variational Inference. Artificial Intelligence and Statistics.
> >
> > M. Titsias. Variational learning of inducing variables in Sparse Gaussian processes. In Artificial Intelligence
> > and Statistics, pages 567–574, 2009.

---

### Review · Reviewer_8g1J · 2024-10-21

**Summary Of Contributions:**

In this work the authors develop a novel version of the stochastic variational gaussian process latent variable model of Lalchand et al which is specifically for multimodal data. Whilst the original model could be applied without adjustment to multimodal data, it would require using the same kernel and hyper-parameters across modalities. If the form of the data is very different across modalities, for example visual and standard continuous covariates, this restriction could lead to poor predictive performance as the same kernel will struggle to capture both. Therefore, the authors develop an approach which is jointly trainable and allows different kernels to be used across modalities. They evaluate their approach on a selection of astrophysics applications for which the methodology was developed.

**Audience:**

Yes

**Broader Impact Concerns:**

No concerns.

**Claims And Evidence:**

Yes

**Requested Changes:**

My requested changes are:
- The notation in section 2 needs to be cleaned up greatly as in its current state it prohibits the full understanding of the technical portion of the paper.
- Add a comparison to a standard predictive approach in the section on predicting properties from spectra.
- Consider toning down or further justifying claims in the conclusion.

**Strengths And Weaknesses:**

I believe the main strengths of the paper are as follows:
1. The paper is a novel extension of Lalchand, and the changes made are well motivated by the goal of fitting a SVI-GPLVM to multimodal data.
2. The authors experimentally evaluate their model in a wide variety of settings that appear strongly motivated by the astrophysics applications of interest. They show the ability of the model to reconstruct from a variety of corrupted samples, as well as the ability of the model to learn to predict scientific labels from quasar spectra alone.

The main weaknesses I find relate to the notation and are as follows:

1. Are $\mathbf{f}_d$ functions or scalars? $\mathbf{f}_d(z_n)$ suggests they are functions being evaluated at $z_n$ but then I find the expressions $\mathcal{N}(\mathbf{f}_d, \cdots)$ confusing as we cannot have a normal distribution over functions, it would be something along the lines of $GP(\mathbf{f}_d, \cdots)$.
2. How does $ K _ {nn} $ depend on  $n$ and what is $n$? In the paper, $ K _ {nn} $ is defined as  $k _ \theta$  evaluated on the set of points $ \mathbf{x}_n$ for $n=1$ to $N$, however $n$ is a dummy variable in the definition so its hard to see its involvement in $ K _ {nn} $. Also, the kernel is defined on $Z$ so should these not be $ \mathbf{z}_n$?
3. Relatedly, I don't see definitions for $K _ {mm} $ or  $K _ {mn}$? I think from the part the play in the posterior I understand what they are but they should be defined.
4. What is the dimension of  $K _ {mm} $? Again from reading between the lines I believe $m=M$ and so this would be an $MxM$ matrix, in which case I do not see how $\mathbf{u}_d$ can have distribution $\mathcal{N}(0,K _ {mm})$ unless $D=M$.
5. In equation (3), why is the final line not $P(X|F,Z,U)$ instead of  $P(X|F,Z)$? The dependence on $Z$ here seems to go.
6. In equation (7), is this the same kernel with different hyper-parameters instead of different kernels with different hyper-parameters? Again the notation make this aspect confusing.
7. For the experiment showing we can predict scientific properties from spectra, how does this approach compare with a more traditional approach that just tries to predict the properties from spectra without learning a latent representation? Strong statements are made on the value of predicting such properties from spectra in the conclusion but it would seem these could be equally well supported by a standard predictive approach as opposed to a multimodal latent variable model.
8. Finally, whilst astrophysics lies outside of my area of expertise I would say that some conclusions in the "Scientific Interpretation and Significance" seem very strong and if more evidence cannot be provided for these claims it would seem they should be toned down slightly. The section lays claim to what seem like two large open astrophysics problems, claiming they are resolved by this method. It would seem to me much more experimental evidence would be required for this claim. However, as stated this is not my area of expertise.

---

> ### Author Response · Authors · 2024-11-03
> **Response to Reviewer 8g1J**
>
> I would like to thank Reviewer 8g1J for their time and thoughtful comments. I enclose the answers to the concerns raised in the review                                                to the best of my ability below:
>
> >1.  Clarifying notation and meaning of $\mathbf{f}\_d$ / $\mathbf{f}\_d(\mathbf{𝑧}\_n)$.
>
> This is a valid concern. The non-bold notation $f\_{d}$ denotes the infinite dimensional function which is GP distributed, like in equation (7) of the paper. The bold notation $\mathbf{f}\_{d}$ denotes the evaluation of the function $f\_d$ on finite inputs in $Z$ so $\mathbf{f}\_{d}$ is a vector of size $N$ given as, $[f\_{d}(\mathbf{z}\_{1}), f\_{d}(\mathbf{z}\_{2}), \ldots, f\_{d}(\mathbf{z}\_{N})]$. Upon careful reading, the notation is incorrect in the 3rd equation of equation block (1) where the mean of the data likelihood should be written as, $f\_{d}(\mathbf{z}\_{n})$ and not $\mathbf{f}\_{d}(\mathbf{z}\_{n})$.
>
> >2.  On $𝐾_{𝑛𝑛}$ and its dependence on $𝑛$ and clarity on inputs.
>
> $K_{nn}$ denotes the kernel matrix of size $N \times N$ evaluated on the inputs using the kernel function $k_{\theta}(\mathbf{z}, \mathbf{z}^{\prime})$; and the inputs under the notation in the paper are $\mathbf{z}\_n$ and not
>  $\mathbf{x}\_n$ that is indeed a typo in just that one location (under eq. block (1)) where I have mistyped the inputs as $\mathbf{x}\_n$. In terms of the notation $K_{nn}$ for the kernel matrix, it is commonly used in GP literature and since this model is based on SVI-GP I deferred to the same notation as in [Hensman et al, 2013, Lalchand et al, 2022].
>
> > 3. Definitions for $K_{mn}$ and $K_{mm}$
>
> Agreed, will define these terms explicitly before introducing them. For clarity, $K\_{mm}$ is the $M \times M$ covariance matrix resulting from evaluating the kernel function on the inducing inputs $\tilde{Z}$ and $K\_{mn}$ is the $M \times N$ cross-covariance matrix resulting from evaluating the kernel fnuction on inducing inputs $\tilde{Z}$ and training inputs $Z$.
>
> > 4. On $K_{mm}$ and distribution of $\mathbf{u}_{d}$.
>
> $K_{mm}$ is indeed a $M \times M$ matrix (this notation is consistent with GP literature, Hensman et al, 2013) and each $\mathbf{u}\_{d}$ ( for every output dimension) is a vector of size $M$ where $M < N$, their prior is indeed $\mathcal{N}(0,K\_{mm})$
> as they are $M$ dimensional vectors. Will add more clarity on this.
>
> >5. In equation (3) why is the factorisation $p(X|F,Z)$ and not $p(X|F,Z,U)$
>
> This arises from the conditional dependency structure of the model, $X$ depends on $F$, and $F$ depends on $U$. Once $F$ is known, the presence of $U$ does not change the conditional distribution of $X$. $U$ influences $X$ only through its effect on $F$. Given $F$, $X$ is conditionally independent of $U$.
>
> >6. On the kernel functions and hyperparameters
>
> We learn two different sets of hypers and choose a squared exponential kernel form for both observation spaces $X$ and $Y$, however, one can also choose different kernel functions depending on the specifics of the modality. For instance, if we wanted to capture periodicity in one of the modalities we could use a periodic kernel function for that space. The hypers are shared within each observation space, we don't learn $D$ sets of hypers, just one set for the pixels and one set for the scientific labels.
>
> > 7. Regarding bechmarking with a more traditional approach.
>
> The dependency between the spectra and labels is modelled through the shared latent space which generates both the views of the data. Without a generative model it would not be possible to generate spectra corresponding to scientific labels (generate $X$ corresponding to $Y$), we would only be able to predict labels based on fixed observed spectra. With a joint generative model it is possible to generate both spectra and labels (jointly) or generate one conditioned only on observing the other observation space. In other words, we can extract the shared latent $Z$ corresponding to $X$ (spectra) in order to predict a reconstructed estimate of the spectra and the labels $(\hat{X},\hat{Y})$ or the other way round, $Y \xrightarrow{optimise} Z \xrightarrow{decoder} (\hat{X}, \hat{Y})$ as shown in experiments. Further, the spectra is only partially observed making regression even harder as one would need to account for missing / uncertain inputs. But, I agree that it is worth emphasizing this point further and including a suitable baseline approach.
>
> > 8. On scientific interpretation and significance
>
> We concur with these concerns and have toned down the whole section, importantly, we have added a short paragraph about further testing at the end of the section.
>
> References:
>
> J. Hensman, N. Fusi, and N. D. Lawrence. Gaussian processes for big data. Proceedings of the 29th Conference on Uncertainty in Artificial Intelligence (UAI), 2013
>
> Vidhi Lalchand, Aditya Ravuri, and Neil D. Lawrence. Generalised GPLVM with Stochastic Variational Inference. AISTATS 2022

---

> > ### Comment · Reviewer_8g1J · 2024-11-21
> > **Response to Authors**
> >
> > Thank you very much to the authors for their clarifications, specifically on the notation. I did not realise this notation follows from [Hensman et al, 2013] and after referring to that paper and the authors comments the notation makes much more sense. However, I do note that in [Hensman et al, 2013] they do define each matrix in the text. I would say for this portion of the paper to be comprehensible, the authors should do the same, ideally declaring the dimension of each vector and matrix when they are first declared alongside their definition.
> >
> > Otherwise, I feel the reviewers response effectively addresses my concerns. I am happy to see both the toning down of the significance and the traditional regression approach as a baseline. As the authors comment, if the missingness in the original spectra $X^*_{gt}$ is enough to create problems for a traditional GP approach, this would provide further justification for the usefulness of their proposed method.

---

> > > ### Author Response · Authors · 2024-11-28
> > > **Response (Round 2)**
> > >
> > > Thank you for your feedback and your approval of the other corrections.
> > >
> > > I have now included the dimensions for all vectors and matrices in the background section when I first introduced them. I have also included a short section about the challenges with benchmarking with a traditional regression model at the end of the experiments and highlighted from the comment above, the precise advantages of using a learnt compression from a partially observed vector to predict other modalities and also impute the missing values. Thank you.

---

### Author Response · Authors · 2024-11-05
**Addressing novelty, technical contribution  and toning down scientific claims [all reviewers]**

**Novelty**

The combination of model, inference and application in this work can be viewed as adding to the line of work on scalable GPLVMs [Lalchand et al, 2022]. They have never been used in conjunction before.

Mathematically, it seems like a simple extension but the sharing of inducing inputs across observation spaces was not trivial to implement. It certainly wasn't clear before running the experiments if the doubly stochastic ELBO for each observation space could be trained in this fashion or if the shared latents would align well enough to represent and reconstruct each observation space. This had to be tested on a concrete application and this is exactly what the paper tries to achieve. Hence, it certainly isn't a trivial extension of known techniques to the best of our knowledge. The proposed method opens up possibilities for various extensions (not considered here), for instance, reweighting of the ELBO terms to fine tune reconstruction performance in both modalities.

**Scientific claims**

To address concerns about the scientific claims, the final section 8 has been re-worded and the paragraph below on the need for further testing has been included.

*Further testing of the required precision with which the quasars' luminosities and thus distances can be predicted by the generative model will be necessary, in order to determine whether the constraints on the universe's expansion rate obtained by the more numerous quasars can improve on the current constraints by the fewer but very accurately measured distances from supernovae.*

---

### Author Response · Authors · 2024-11-06
**Changes in current / future revision summarised**

Dear Reviewers,

The revision uploaded has most of the minor changes / typos / equations and strucural chanes addressed as well as a rewording of the scientific interpretation section.

Below, I summarise the additional experiments which I aim to include in a future revision (although, I am unsure how much time I have to do so before the decision). All of them seem very reasonable and within the scope so I don't forsee any specific challenges in including them in the manuscript.

1. A traditional regression experiment, predicting scientific labels from spectra, this approach would have to adapt for missing dimensions in the spectra. [Reviewer 8g1J]
2. Baseline model for Table 2 uncertainty quantification experiments.[Reviewer T1nP]
3. Masked autoencoder type benchmark model adapted to multi-view setting for reconstruction accuracy comparison.[Reviewer T1nP]
4. A calibration plot (ground truth vs. predictions in percentile buckets) to understand prediction quality for the labels.[Reviewer qTSY ]
5. A plot for error vs. SNR to establish support for the statement that they are uncorrelated. [Reviewer qTSY]
6. Plots resolution and fix '2\sigma' labels to 1.96\sigma or 95% prediction interval. [Reviewer qTSY]

---

### Author Response · Authors · 2024-11-30
**Final Revision uploaded**

Dear Reviewers,

Below is a summary of additional experiments in the final revision which is now uploaded.

1. SNR vs error plots, analysis  [Appendix D][Reviewer qTSY]
2. Table 2 NLPD estimates with SE across data splits for the scientific labels. [Page 13] [Reviewer T1nP]
3. Disentanglement version of the model with individual hyperparameters for each scientific label added in Table 1. [Page 9] [Reviewer qTSY].
4. Calibration plot Error vs. Uncertainty [Appendix E][Reviewer qTSY]

Things I have noted to include in the final version post decision:

1. Sharpening of the plots and correction of the legend.

2 additional scripts carrying the code for this experiment are also added to the supplementary file.

disentangled_gplvm.py

disentangled_gplvm_run.py

---

### Decision · Action_Editor_m737 · 2024-12-06

**Recommendation:** Accept as is

**Comment:**

All three reviewers agree that the final version deserves to be accepted. I share their opinion, especially with the additional experimental evaluations and improvements in the text. As all these have already been included in the current version a minor revision is not necessary and I recommend acceptance.

**Audience:**

The paper's main focus is the application area of astrophysics, especially quasar spectra instead of providing deep theoretical advances.
As such its primary audience is the subset of astrophysicists within the TMLR readership as well as anybody who is interested in real-world applications of Gaussian processes latent-variable models.

**Claims And Evidence:**

After some reviewers had some doubts about the experimental evaluation at submission time, the additional experiments added throughout the discussion period do now provide a sufficient amount of evidence to validate the authors' claims.